# Neural structure learning with stochastic differential equations

**Benjie Wang**[*†‡]
University of California, Los Angeles

**Joel Jennings**
DeepMind

**Wenbo Gong**[*†]
Microsoft Research Cambridge

## Abstract

Discovering the underlying relationships among variables from temporal observations has been a longstanding challenge in numerous scientific disciplines, including biology, finance, and climate science. The dynamics of such systems are often best described using continuous-time stochastic processes. Unfortunately, most existing structure learning approaches assume that the underlying process evolves in discrete-time and/or observations occur at regular time intervals. These mismatched assumptions can often lead to incorrect learned structures and models. In this work, we introduce a novel structure learning method, *SCOTCH*, which combines neural stochastic differential equations (SDE) with variational inference to infer a posterior distribution over possible structures. This continuous-time approach can naturally handle both learning from and predicting observations at arbitrary time points. Theoretically, we establish sufficient conditions for an SDE and *SCOTCH* to be structurally identifiable, and prove its consistency under infinite data limits. Empirically, we demonstrate that our approach leads to improved structure learning performance on both synthetic and real-world datasets compared to relevant baselines under regular and irregular sampling intervals.

## 1 Introduction

Time-series data is ubiquitous in the real world, often comprising a series of data points recorded at varying time intervals. Understanding the underlying structures between variables associated with temporal processes is of paramount importance for numerous real-world applications (Spirtes et al., 2000; Berzuini et al., 2012; Peters et al., 2017). Although randomised experiments are considered the gold standard for unveiling such relationships, they are frequently hindered by factors such as cost and ethical concerns. Structure learning seeks to infer hidden structures from purely observational data, offering a powerful approach for a wide array of applications (Bellot et al., 2022; Löwe et al., 2022; Runge, 2018; Tank et al., 2021; Pamfil et al., 2020; Gong et al., 2022).

However, many existing structure learning methods for time series are discrete, assuming that the underlying temporal processes are discretized in time and requiring uniform sampling intervals. Consequently, these models face two limitations: (i) they may misrepresent the true underlying process when it is continuous, potentially leading to incorrect inferred relationships; and (ii) they struggle with handling irregular sampling intervals, which frequently arise in many fields (Trapnell et al., 2014; Qiu et al., 2017; Qian et al., 2020) and climate science (Bracco et al., 2018; Raia, 2008).

To address these challenges, we introduce a novel framework, **S**tructure learning with **CO**ntinuous-**T**ime sto**CH**astic models (*SCOTCH*[1]), which employs *stochastic differential equations* (SDEs) for structure learning in temporal processes. *SCOTCH* can naturally handle irregularly sampled time series and accurately represent and learn continuous-time processes. We make the key contributions:

1. We introduce a novel latent Stochastic Differential Equation (SDE) formulation for modelling structure in continuous-time observational time-series data. To effectively train our

---

[*]Equal contribution.

[†]Correspondence to `benjiewang@ucla.edu, wenbogong@microsoft.com`

[‡]Work done during an internship at Microsoft Research Cambridge.

[1]`https://github.com/microsoft/causica/tree/main/research_experiments/scotch`

proposed model, which we denote as *SCOTCH*, we adapt the variational inference framework proposed in (Li et al., 2020; Tzen & Raginsky, 2019a) to approximate the posterior for both the underlying graph structure and the latent variables. We show that, in contrast to a prior approach using ordinary differential equations (ODEs) (Bellot et al., 2022), our model is capable of accurately learning the underlying dynamics from trajectories exhibiting multimodality and with non-Gaussian distribution.

2. We provide a rigorous theoretical analysis to support our proposed methodology. Specifically, we prove that when SDEs are directly employed for modelling the observational process, the resulting SDEs are structurally identifiable under global Lipschitz and diagonal noise assumptions. We also prove our model maintains structural identifiability under certain conditions, even when adopting the latent formulation; and that variational inference, when integrated with the latent formulation, in the infinite data limit, can successfully recover the ground truth graph structure and mechanisms under specific assumptions.

3. Empirically, we conduct extensive experiments on both synthetic and real-world datasets showing that *SCOTCH* can improve upon existing methods on structure learning, including when the data is irregularly sampled.

## 2 PRELIMINARIES

In the rest of this paper, we use $\boldsymbol{X}_t \in \mathbb{R}^D$ to denote the $D$-dimensional observation vector at time $t$, with $X_{t,d}$ representing the $d^{\text{th}}$ variable of the observation. A time series is a set of $I$ observations $\boldsymbol{X} = \{\boldsymbol{X}_{t_i}\}_{t=1}^I$, where $\{t_i\}_{i=1}^I$ are the observation times. In the case where we have multiple ($N$) i.i.d. time series, we use $\boldsymbol{X}^{(n)}$ to indicate the $n^{\text{th}}$ time series.

**Bayesian structure learning**    In structure learning, the aim is to infer the underlying graph representing the relationships between variables from data. In the Bayesian approach, given time series data $\{\boldsymbol{X}^{(n)}\}_{n=1}^N$, we define a joint distribution over graphs and data given by:

$$p(\boldsymbol{G}, \boldsymbol{X}^{(1)}, \ldots, \boldsymbol{X}^{(N)}) = p(\boldsymbol{G}) \prod_{n=1}^N p(\boldsymbol{X}^{(n)}|\boldsymbol{G}) \tag{1}$$

where $p(\boldsymbol{G})$ is the graph prior and $p(\boldsymbol{X}^{(n)}|\boldsymbol{G})$ is the likelihood term. The goal is then to compute the graph posterior $p(\boldsymbol{G}|\boldsymbol{X}^{(1)}, \ldots \boldsymbol{X}^{(N)})$. However, analytic computation is intractable in high dimensional settings. Therefore, variational inference (Zhang et al., 2018) and sampling methods (Welling & Teh, 2011; Gong et al., 2018; Annadani et al., 2023) are commonly used for inference.

**Structural equation models (SEMs)**    Given a time series $\boldsymbol{X}$ and graph $\boldsymbol{G} \in \{0,1\}^{D \times D}$, we can use SEMs to describe the structural relationships between variables:

$$X_{t,d} = f_{t,d}(\boldsymbol{Pa_G}^d(< t), \epsilon_{t,d}) \tag{2}$$

where $\boldsymbol{Pa_G}^d(< t)$ specifies the lagged parents of $X_{t,d}$ at previous time and $\epsilon_{t,d}$ is the mutually independent noise. Such a model requires discrete time steps that are usually assumed to follow a regular sampling interval, i.e. $t_{i+1} - t_i$ is a constant for all $i = 1, \ldots, I - 1$. Most existing models can be regarded as a special case of this framework.

**Itô diffusion**    A time-homogenous Itô diffusion is a stochastic process $\boldsymbol{X}_t$ and has the form:

$$d\boldsymbol{X}_t = \boldsymbol{f}(\boldsymbol{X}_t)dt + \boldsymbol{g}(\boldsymbol{X}_t)d\boldsymbol{W}_t \tag{3}$$

where $\boldsymbol{f} : \mathbb{R}^D \to \mathbb{R}^D, \boldsymbol{g} : \mathbb{R}^D \to \mathbb{R}^{D \times D}$ are time-homogeneous drift and diffusion functions, respectively, and $\boldsymbol{W}_t$ is a Brownian motion under the measure $P$. It is known that under global Lipschitz guarantees (Assumption 1) this has a unique strong solution (Øksendal & Øksendal, 2003).

**Euler discretization and Euler SEM**    For most Itô diffusions, the analytic solution $\boldsymbol{X}_t$ is intractable, especially with non-linear drift and diffusion functions. Thus, we often seek to simulate

the trajectory by discretization. One common discretization method is the *Euler-Maruyama* (EM) scheme. With a fixed step size $\Delta$, EM simulates the trajectory as

$$\boldsymbol{X}_{t+1}^{\Delta} = \boldsymbol{X}_t^{\Delta} + \boldsymbol{f}(\boldsymbol{X}_t^{\Delta})\Delta + \boldsymbol{g}(\boldsymbol{X}_t^{\Delta})\eta_t \tag{4}$$

where $\boldsymbol{X}_t^{\Delta}$ is the random variable induced by discretization and $\eta_t \sim \mathcal{N}(0, \Delta)$. Notice that eq. (4) is a special case of eq. (2). If we define the graph $\boldsymbol{G}$ as the following: $\boldsymbol{X}_{t,i}^{\Delta} \to \boldsymbol{X}_{t+1,j}^{\Delta}$ in $\boldsymbol{G}$ iff $\frac{\partial f_j(\boldsymbol{X}_t^{\Delta})}{\partial X_{t,i}^{\Delta}} \neq 0$ or $\exists k, \frac{\partial g_{j,k}(\boldsymbol{X}_t^{\Delta})}{\partial X_{t,i}^{\Delta}} \neq 0$; and assume $\boldsymbol{g}$ only outputs a diagonal matrix, then the above EM induces a temporal SEM, called the *Euler SEM* (Hansen & Sokol, 2014), which provides a useful analysis tool for continuous time processes.

# 3 *SCOTCH*: BAYESIAN STRUCTURE LEARNING FOR CONTINUOUS TIME SERIES

We consider a dynamical system in which there is both intrinsic stochasticity in the evolution of the state, as well as independent measurement noise that is present in the observed data. For example, in healthcare, the condition of a patient will progress with randomness rather than deterministically. Further, the measurement of patient condition will also be affected by the accuracy of the equipment, where the noise is independent to the intrinsic stochasticity. To account for this behaviour, we propose to use the latent SDE formulation (Li et al., 2020; Tzen & Raginsky, 2019a):

$$d\boldsymbol{Z}_t = \boldsymbol{f}_\theta(\boldsymbol{Z}_t)dt + \boldsymbol{g}_\theta(\boldsymbol{Z}_t)d\boldsymbol{W}_t \text{ (latent process)}$$
$$\boldsymbol{X}_t = \boldsymbol{Z}_t + \boldsymbol{\epsilon}_t \text{ (noisy observations)} \tag{5}$$

where $\boldsymbol{Z}_t \in \mathbb{R}^D$ is the latent variable representing the internal state of the dynamic system, $\boldsymbol{X}_t \in \mathbb{R}^D$ describes the observational data with the same dimension, $\boldsymbol{\epsilon}_t$ is additive dimension-wise independent noise, $\boldsymbol{f}_\theta : \mathbb{R}^D \to \mathbb{R}^D$ is the drift function, $\boldsymbol{g}_\theta : \mathbb{R}^D \to \mathbb{R}^{D \times D}$ is the diffusion function and $\boldsymbol{W}_t$ is the Wiener process. For *SCOTCH*, we require the following two assumptions:

**Assumption 1** (Global Lipschitz). *We assume that the drift and diffusion functions in eq. (5) satisfy the global Lipschitz constraints. Namely, we have*

$$|\boldsymbol{f}_\theta(\boldsymbol{x}) - \boldsymbol{f}_\theta(\boldsymbol{y})| + |\boldsymbol{g}_\theta(\boldsymbol{x}) - \boldsymbol{g}_\theta(\boldsymbol{y})| \leq C|\boldsymbol{x} - \boldsymbol{y}| \tag{6}$$

*for some constant $C$, $\boldsymbol{x}, \boldsymbol{y} \in \mathbb{R}^D$ and $|\cdot|$ is the corresponding $L_2$ norm for vector-valued functions and matrix norm for matrix-valued functions.*

**Assumption 2** (Diagonal diffusion). *We assume that the diffusion function $\boldsymbol{g}_\theta$ outputs a non-zero diagonal matrix. That is, it can be simplified to a vector-valued function $\boldsymbol{g}_\theta(\boldsymbol{X}_t) : \mathbb{R}^D \to \mathbb{R}^D$.*

The former is a standard assumption required by most SDE literature to ensure the existence of a strong solution. The key distinction is the latter assumption of a nonzero diagonal diffusion function, $\boldsymbol{g}_\theta$, rather than a full diffusion matrix, enabling structural identifiability as we show in the next section. Please refer to appendix A.1 for more detailed explanations. Now, in accordance with the graph defined in Euler SEMs (section 2), we define the *signature graph $\boldsymbol{G}$* as follows: edge $i \to j$ is present in $\boldsymbol{G}$ iff $\exists t$ s.t. either $\frac{\partial f_j(\boldsymbol{Z}_t)}{\partial Z_{t,i}} \neq 0$ or $\frac{\partial g_j(\boldsymbol{Z}_t)}{\partial Z_{t,i}} \neq 0$. Note that there is no requirement for the graph to be acyclic. Intuitively, the graph $\boldsymbol{G}$ describes the structural dependence between variables.

We now instantiate our Bayesian structure learning scheme with prior and likelihood components:

**Prior over Graphs** Leveraging Geffner et al. (2022); Annadani et al. (2023), our graph prior is designed as:

$$p(\boldsymbol{G}) \propto \exp(-\lambda_s \|\boldsymbol{G}\|_F^2) \tag{7}$$

where $\lambda_s$ is the graph sparsity coefficient, and $\|\cdot\|_F$ is the Frobenius norm.

**Prior process** Since the latent process induces a distribution over latent trajectories $p_\theta(\boldsymbol{Z})$ before seeing any observations, we also call it the prior process. We propose to use neural networks for drift and diffusion functions $\boldsymbol{f}_\theta : \mathbb{R}^D \times \{0, 1\}^{D \times D} \to \mathbb{R}^D$, $\boldsymbol{g}_\theta : \mathbb{R}^D \times \{0, 1\}^{D \times D} \to \mathbb{R}^D$, which explicitly take the latent state and the graph as inputs. Though the signature graph is defined through

the function derivatives, we explicitly use the graph $\boldsymbol{G}$ as input to enforce the constraint. We will interchangeably use the notation $\boldsymbol{f}_G$ and $\boldsymbol{g}_G$ to denote $\boldsymbol{f}_\theta(\cdot, \boldsymbol{G})$ and $\boldsymbol{g}_\theta(\cdot, \boldsymbol{G})$. For the graph-dependent drift and diffusion, we leverage the design of Geffner et al. (2022) and propose:

$$\boldsymbol{f}_{G,d}(\boldsymbol{Z}_t) = \zeta\left(\sum_{i=1}^{D} G_{i,d} l(Z_{t,i}, \boldsymbol{e}_i), \boldsymbol{e}_d\right) \tag{8}$$

for both $\boldsymbol{f}_G$ and $\boldsymbol{g}_G$, where $\zeta$, $l$ are neural networks, and $\boldsymbol{e}_i$ is a trainable node embedding for the $i^{\text{th}}$ node. The corresponding prior process is:

$$d\boldsymbol{Z}_t = \boldsymbol{f}_\theta(\boldsymbol{Z}_t, \boldsymbol{G})dt + \boldsymbol{g}_\theta(\boldsymbol{Z}_t, \boldsymbol{G})d\boldsymbol{W}_t \text{ (prior process)} \tag{9}$$

**Likelihood of time series** Given a time series $\boldsymbol{X} = \{\boldsymbol{X}_{t_i}\}_{i=1}^{I}$, the likelihood is defined as

$$p(\{\boldsymbol{X}_{t_i}\}_{i=1}^{I}|\{\boldsymbol{Z}_{t_i}\}_{i=1}^{I}, \boldsymbol{G}) = \prod_{i=1}^{I}\prod_{d=1}^{D} p_{\epsilon_d}(X_{t_i,d} - Z_{t_i,d}) \tag{10}$$

where $p_{\epsilon_d}$ is the observational noise distribution for the $d^{\text{th}}$ dimension.

## 3.1 VARIATIONAL INFERENCE

Suppose that we are given multiple time series $\{\boldsymbol{X}^{(n)}\}_{n=1}^{N}$ as observed data from the system. The goal is then to compute the posterior over graph structures $p(\boldsymbol{G}|\{\boldsymbol{X}^{(n)}\}_{n=1}^{N})$, which is intractable. Thus, we leverage variational inference to simultaneously approximate both the graph posterior, and a latent posterior process over $\boldsymbol{Z}^{(n)}$ for every observed time series $\boldsymbol{X}^{(n)}$.

Given $N$ i.i.d time series $\{\boldsymbol{X}^{(n)}\}_{n=1}^{N}$, we propose to use a variational approximation $q_\phi(\boldsymbol{G}) \approx p(\boldsymbol{G}|\boldsymbol{X}^{(1)}, \ldots, \boldsymbol{X}^{(N)})$. With the standard trick from variational inference, we have the following evidence lower bound (ELBO):

$$\log p(\boldsymbol{X}^{(1)}, \ldots, \boldsymbol{X}^{(N)}) \geq \mathbb{E}_{q_\phi(\boldsymbol{G})}\left[\sum_{n=1}^{N} \log p_\theta(\boldsymbol{X}^{(n)}|\boldsymbol{G})\right] - D_{\text{KL}}(q_\phi(\boldsymbol{G})\|p(\boldsymbol{G})) \tag{11}$$

Unfortunately, the distribution $p_\theta(\boldsymbol{X}^{(n)}|\boldsymbol{G})$ remains intractable due to the marginalization of the latent Itô diffusion $\boldsymbol{Z}^{(n)}$. Therefore, we leverage the variational framework proposed in Tzen & Raginsky (2019a); Li et al. (2020) to approximate the true posterior $p(\boldsymbol{Z}^{(n)}|\boldsymbol{X}^{(n)}, \boldsymbol{G})$. For each $n = 1, \ldots, N$, the variational posterior $q_\psi(\tilde{\boldsymbol{Z}}^{(n)}|\boldsymbol{X}^{(n)}, \boldsymbol{G})$ is defined as the solution to the following:

$$\tilde{\boldsymbol{Z}}_{t,0}^{(n)} \sim \mathcal{N}(\boldsymbol{\mu}_\psi(\boldsymbol{G}, \boldsymbol{X}^{(n)}), \boldsymbol{\Sigma}_\psi(\boldsymbol{G}, \boldsymbol{X}^{(n)})) \text{ (posterior initial state)}$$

$$d\tilde{\boldsymbol{Z}}_t^{(n)} = \boldsymbol{h}_\psi(\tilde{\boldsymbol{Z}}_t^{(n)}, t; \boldsymbol{G}, \boldsymbol{X}^{(n)})dt + \boldsymbol{g}_G(\tilde{\boldsymbol{Z}}_t^{(n)})d\boldsymbol{W}_t \text{ (posterior process)} \tag{12}$$

For the initial latent state, $\boldsymbol{\mu}_\psi, \boldsymbol{\Sigma}_\psi$ are posterior mean and covariance functions implemented as neural networks. For the SDE, we use the same diffusion function $\boldsymbol{g}_G$ for both the prior and posterior processes, but train a separate neural drift function $\boldsymbol{h}_\psi$ for the posterior, which takes a time series $\boldsymbol{X}^{(n)}$ as input. The posterior drift function differs from the prior in two key ways. Firstly, the posterior drift function depends on time; this is necessary as conditioning on the observed data creates this dependence even when the prior process is time-homogenous. Secondly, while $\boldsymbol{h}_\psi$ takes the graph $\boldsymbol{G}$ as an input, the function design is not constrained to have a matching signature graph like $\boldsymbol{f}_G$. More details on the implementation of $\boldsymbol{h}_\psi, \boldsymbol{\mu}_\psi, \boldsymbol{\Sigma}_\psi$ can be found in Appendix B.

Assume for each time series $\boldsymbol{X}^{(n)}$, we have observation times $t_i$ for $i = 1, \ldots, I$ within the time range $[0, T]$, then, we have the following evidence lower bound for $\log p(\boldsymbol{X}^{(n)}|\boldsymbol{G})$ (Li et al., 2020):

$$\log p(\boldsymbol{X}^{(n)}|\boldsymbol{G}) \geq \mathbb{E}_{q_\psi}\left[\sum_{i=1}^{I} \log p(\boldsymbol{X}_{t_i}^{(n)}|\tilde{\boldsymbol{Z}}_{t_i}^{(n)}, \boldsymbol{G}) - \int_0^T \|\boldsymbol{u}^{(n)}(\tilde{\boldsymbol{Z}}_t^{(n)})\|^2 dt\right] \tag{13}$$

where $\tilde{\boldsymbol{Z}}_t^{(n)}$ is the posterior process modelled by eq. (12) and $\boldsymbol{u}^{(n)}(\tilde{\boldsymbol{Z}}_t^{(n)})$ is given by:

$$\boldsymbol{u}^{(n)}(\tilde{\boldsymbol{Z}}_t^{(n)}) = \boldsymbol{g}_G(\tilde{\boldsymbol{Z}}_t^{(n)})^{-1}(\boldsymbol{h}_\psi(\tilde{\boldsymbol{Z}}_t^{(n)}, t; \boldsymbol{G}, \boldsymbol{X}^{(n)}) - \boldsymbol{f}_G(\tilde{\boldsymbol{Z}}_t^{(n)})) \tag{14}$$

---

**Algorithm 1** *SCOTCH* training

---

**Input:** i.i.d time series $\{\boldsymbol{X}^{(n)}\}_{n=1}^N$; drift functions $\boldsymbol{f}_G$, $\boldsymbol{h}_\psi$, diffusion function $\boldsymbol{g}_G$, SDE solver Solver, initial condition $\tilde{\boldsymbol{Z}}_0^{(n)}$, training iterations L
**for** $l = 1, \ldots, L$ **do**
    Sample time series mini-batch $\{\boldsymbol{X}^{(n)}\}_{n=1}^S$ with batch size $S$.
    **for** $n = 1, \ldots, S$ **do**
        Draw graph $\boldsymbol{G} \sim q_\phi(\boldsymbol{G})$
        Draw initial latent state $\tilde{\boldsymbol{Z}}_0^{(n)} \sim \mathcal{N}(\boldsymbol{\mu}_\psi(\boldsymbol{G}, \boldsymbol{X}^{(n)}), \boldsymbol{\Sigma}_\psi(\boldsymbol{G}, \boldsymbol{X}^{(n)}))$
        Solve (sample from) the posterior process $(\tilde{\boldsymbol{Z}}^{(n)}, L) = \text{Solver}((\tilde{\boldsymbol{Z}}_0^{(n)}, 0), \boldsymbol{f}_G, \boldsymbol{h}_\psi, \boldsymbol{g}_G)$
    **end for**
    Maximize ELBO eq. (15) w.r.t. $\phi, \psi, \theta$
**end for**

---

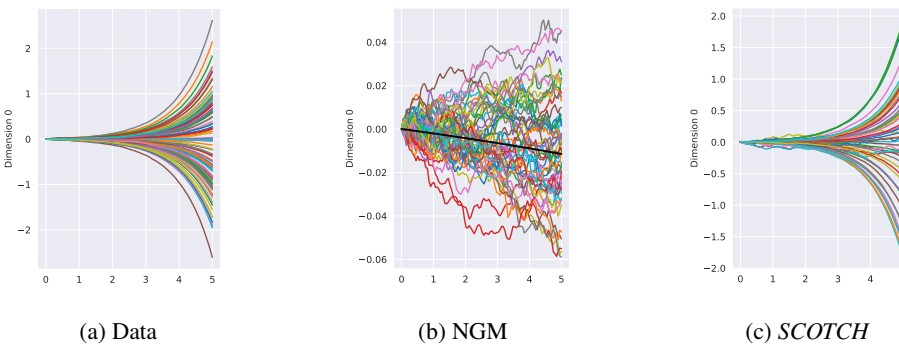

(a) Data          (b) NGM          (c) *SCOTCH*

Figure 1: Comparison between NGM and *SCOTCH* for simple SDE (note vertical axis scale)

By combining eq. (11) and eq. (13), we derive an overall ELBO:

$$\log p_\theta(\boldsymbol{X}^{(1)}, \ldots, \boldsymbol{X}^{(N)}) \geq \mathbb{E}_{q_\phi}\left[\sum_{n=1}^N \mathbb{E}_{q_\psi}\left[\sum_{i=1}^I \log p(\boldsymbol{X}_{t_i}^{(n)}|\tilde{\boldsymbol{Z}}_{t_i}^{(n)}, \boldsymbol{G}) - \int_0^T \|\boldsymbol{u}^{(n)}(\tilde{\boldsymbol{Z}}_t^{(n)})\|^2 dt\right]\right]$$
$$- D_{\mathrm{KL}}(q_\phi(\boldsymbol{G})\|p(\boldsymbol{G})) \tag{15}$$

In practice, we approximate the ELBO (and its gradients) using a Monte-Carlo approximation. The inner expectation can be approximated by simulating from an augmented version of eq. (12) where an extra variable $L$ is added with drift $\frac{1}{2}|\boldsymbol{u}^{(n)}(\tilde{\boldsymbol{Z}}_t^{(n)})|^2$ and diffusion zero (Li et al., 2020). Algorithm 1 summarizes the training algorithm of *SCOTCH*.

### 3.2 Stochasticity and Continuous-Time Modeling

**Stochasticity** Bellot et al. (2022) proposed a structure learning method, called NGM, to learn from single time series generated by SDEs. NGM uses a neural ODE to model the mean process $\boldsymbol{f}_\theta$, and extracts graphical structure from the first layer of $\boldsymbol{f}_\theta$. However, NGM assumes that the observed single series $\boldsymbol{X}$ follows a multivariate Gaussian distribution, which only holds for linear SDEs. If this assumption is violated, optimizing their proposed squared loss cannot recover the underlying system. *SCOTCH* does not have this limitation and can handle more flexible state-dependent drifts and diffusions. Another drawback of NGM is its inability to handle multiple time series ($N > 1$). Learning from multiple series is important when dealing with SDEs with multimodal behaviour. We propose a simple bimodal 1-D failure case: $dX = Xdt + 0.01dW_t, X_0 = 0$, with the signature graph containing a self-loop. Figure 1 shows the bimodal trajectories (upwards and downwards) sampled from the SDE. The optimal ODE mean process in this case is the constant $\boldsymbol{f}_\theta = 0$ with an empty graph, as confirmed by the learned mean process of NGM (black line in fig. 1b). In contrast, *SCOTCH* can learn the underlying SDE and simulate the correct trajectories (fig. 1c).

**Discrete vs Continuous-Time** Gong et al. (2022) proposed a flexible discretised temporal SEM, called Rhino, that is capable of modelling (1) lagged parents; (2) instantaneous effect; and (3) history

dependent noise. Rhino's SEM is given by $X_{t,d} = f_d(\boldsymbol{Pa_G}^d(< t), \boldsymbol{Pa_G}^d(t)) + g_d(\boldsymbol{Pa_G}^d(< t))\epsilon_{t,d}$. We can clearly see its similarity to *SCOTCH*. If $f_d$ has a residual structure as $f_d(\cdot) = X_{t,d} + r_d(\cdot)\Delta$ and we assume no instantaneous effect ($\boldsymbol{Pa_G}^d(t)$ is empty), Rhino SEM is equivalent to the Euler SEM of the latent process (eq. (9)) with drift $\boldsymbol{r}$, step size $\Delta$ and diagonal diffusion $\boldsymbol{g}$. Thus, similar to the relation of ResNet (He et al., 2016) to NeuralODE (Chen et al., 2018), *SCOTCH* can be interpreted as the continuous-time analog of Rhino.

## 4 THEORETICAL CONSIDERATIONS OF *SCOTCH*

In this section, we aim to answer three important theoretical questions regarding the Itô diffusion proposed in section 3. For notational simplicity, we consider the single time series setting. First, we examine when a general Itô diffusion is structurally identifiable. Secondly, we consider structural identifiability in the latent formulation of eq. (5). Finally, we consider whether optimising ELBO (eq. (15)) can recover the true graph and mechanism if we have infinite observations of a single time series within a fixed time range $[0, T]$. All detailed proofs, definitions, and assumptions can be found in appendix A.

### 4.1 STRUCTURE IDENTIFIABILITY

Suppose that the observational process is given as an Itô diffusion:

$$d\boldsymbol{X}_t = \boldsymbol{f}_G(\boldsymbol{X}_t)dt + \boldsymbol{g}_G(\boldsymbol{X}_t)d\boldsymbol{W}_t \tag{16}$$

Then, we might ask what are sufficient conditions for the model to be structurally identifiable? That is, there does not exist $\boldsymbol{G}' \neq \boldsymbol{G}$ that can induce the same observational distribution.

**Theorem 4.1** (Structure identifiability of the observational process). *Given eq. (16), let us define another process with $\bar{\boldsymbol{X}}_t$, $\boldsymbol{G} \neq \bar{\boldsymbol{G}}$, $\bar{\boldsymbol{f}}_{\bar{G}}$, $\bar{\boldsymbol{g}}_{\bar{G}}$ and $\bar{\boldsymbol{W}}_t$. Then, under Assumptions 1-2, and with the same initial condition $\boldsymbol{X}(0) = \bar{\boldsymbol{X}}(0) = \boldsymbol{x}_0$, the solutions $\boldsymbol{X}_t$ and $\bar{\boldsymbol{X}}_t$ will have different distributions.*

Next, we show that structural identifiability is preserved, under certain conditions, even in the latent formulation where the SDE solution is not directly observed.

**Theorem 4.2** (Structural identifiability with latent formulation). *Consider the distributions $p, \bar{p}$ defined by the latent model in eq. (5) with $(\boldsymbol{G}, \boldsymbol{Z}, \boldsymbol{X}, \boldsymbol{f}_G, \boldsymbol{g}_G), (\bar{\boldsymbol{G}}, \bar{\boldsymbol{Z}}, \bar{\boldsymbol{X}}, \bar{\boldsymbol{f}}_{\bar{G}}, \bar{\boldsymbol{g}}_{\bar{G}})$ respectively, where $\boldsymbol{G} \neq \bar{\boldsymbol{G}}$. Further, let $t_1, \ldots, t_I$ be the observation times. Then, under Assumptions 1 and 2:*

1. *if $t_{i+1} - t_i = \Delta$ for all $i \in 1, ..., I - 1$, then $p^\Delta(\boldsymbol{X}_{t_1}, \ldots, \boldsymbol{X}_{t_I}) \neq \bar{p}^\Delta(\bar{\boldsymbol{X}}_{t_1}, \ldots, \bar{\boldsymbol{X}}_{t_I})$, where $p^\Delta$ is the density generated by the Euler discretized eq. (9) for $\boldsymbol{Z}_t$;*

2. *if we have a fixed time range $[0, T]$, then the path probability $p(\boldsymbol{X}_{t_1}, \ldots, \boldsymbol{X}_{t_I}) \neq \bar{p}(\bar{\boldsymbol{X}}_{t_1}, \ldots, \bar{\boldsymbol{X}}_{t_I})$ under the limit of infinite data ($I \to \infty$).*

### 4.2 CONSISTENCY

Building upon the structural identifiability, we can prove the consistency of the variational formulation. Namely, in the infinite data limit, one can recover the ground truth graph and mechanism by maximizing ELBO with a sufficiently expressive posterior process and a correctly specified model.

**Theorem 4.3** (Consistency of variational formulation). *Suppose Assumptions 1-4 are satisfied for the latent formulation (eq. (5)). Then, for a fixed observation time range $[0, T]$, as the number of observations $I \to \infty$, when ELBO (eq. (15)) is maximised, $q_\phi(\boldsymbol{G}) = \delta(\boldsymbol{G}^*)$, where $\boldsymbol{G}^*$ is the ground truth graph, and the latent formulation recovers the underlying ground truth mechanism.*

## 5 RELATED WORK

**Discrete time causal models** The majority of the existing approaches are inherently discrete in time. Assaad et al. (2022) provides a comprehensive overview. There are three types of discovery methods: (1) Granger causality; (2) structure equation model (SEM); and (3) constraint-based

methods. Granger causality assumes that no instantaneous effects are present and the causal direction cannot flow backward in time. Wu et al. (2020); Shojaie & Michailidis (2010); Siggiridou & Kugiumtzis (2015); Amornbunchornvej et al. (2019) leverage the vector-autoregressive model to predict future observations. Löwe et al. (2022); Tank et al. (2021); Bussmann et al. (2021); Dang et al. (2019); Xu et al. (2019); Khanna & Tan (2019) utilise deep neural networks for prediction. Recently, Cheng et al. (2023) introduced a deep-learning based Granger causality that can handle irregularly sampled data, treating it as a missing data problem and proposing a joint framework for data imputation and graph fitting. SEM based approaches assume an explicit causal model associated to the temporal process. Hyvärinen et al. (2010) leverages the identifiability of additive noise models (Hoyer et al., 2008) to build a linear auto-regressive SEM with non-Gaussian noise. Pamfil et al. (2020) utilises the NOTEARS framework (Zheng et al., 2018) to continuously relax the DAG constraints for fully differentiable structure learning. The recently proposed Gong et al. (2022) extended the prior DECI Geffner et al. (2022) framework to handle time series data and is capable of modelling instantaneous effect and history-dependent noise. Constraint-based approaches use conditional independence tests to determine the causal structures. Runge et al. (2019) combines the PC (Spirtes et al., 2000) and momentary conditional independence tests for the lagged parents. PCMCI+ (Runge, 2020) can additionally detect the instantaneous effect. LPCMCI (Reiser, 2022) can further handle latent confounders. CD-NOD (Zhang et al., 2017) is designed to handle non-stationary heterogeneous time series data. However, all constraint-based approaches can only identify the graph up to Markov equivalence class without the functional relationship between variables.

**Continuous time causal models** In terms of using differential equations to model the continuous temporal process, Hansen & Sokol (2014) proposed using stochastic differential equations to describe the temporal causal system. They proved identifiability with respect to the intervention distributions, but did not show how to learn a corresponding SDE. Penalised regression has been explored for linear models, where parameter consistency has been established (Ramsay et al., 2007; Chen et al., 2017; Wu et al., 2014). Recently, NGM (Bellot et al., 2022) uses ODEs to model the temporal process with both identifiability and consistency results. As discussed in previous sections, *SCOTCH* is based on SDEs rather than ODEs, and can model the intrinsic stochasticity within the causal system, whereas NGM assumes deterministic state transitions.

## 6 EXPERIMENTS

**Baselines and Metrics** We benchmark our method against a representative sample of baselines: (i) VARLiNGaM (Hyvärinen et al., 2010), a linear SEM based approach; (ii) PCMCI+ (Runge, 2018; 2020), a constraint-based method for time series; (iii) CUTS, a Granger causality approach which can handle irregular time series; (iv) Rhino (Gong et al., 2022), a non-linear SEM based approach with history-dependent noise and instantaneous effects; and (v) NGM (Bellot et al., 2022), a continuous-time ODE based structure learner. Since most methods require a threshold to determine the graph, we use the threshold-free *area under the ROC curve* (AUROC) as the performance metric. In appendix D, we also report F1 score, true positive rate (TPR) and false discovery rate (FDR).

**Setup** Both the synthetic datasets (Lorenz-96, Glycolysis) and real-world datasets (DREAM3, Netsim) consist of multiple time series. However, it is not trivial to modify NGM and CUTS to support multiple time series. For fair comparison, we use the concatenation of multiple time series, which we found empirically to improve performance. We also mimic irregularly sampled data by randomly dropping observations, which VARLiNGaM, PCMCI, and Rhino cannot handle; in these cases, for these methods we impute the missing data using zero-order hold (ZOH). For *SCOTCH*, we use pathwise gradient estimators with Euler discretization for solving the SDE (see appendix D.1 for discussion on this choice). Further experimental details can be found in Appendices B, C, D.

### 6.1 SYNTHETIC EXPERIMENTS: LORENZ AND GLYCOLYSIS

First, we evaluate *SCOTCH* on synthetic benchmarks including the Lorenz-96 (Lorenz, 1996) and Glycolysis (Daniels & Nemenman, 2015) datasets, which model continuous-time dynamical systems. The Lorenz model is a well-known example of chaotic systems observed in biology (Goldberger & West, 1987; Heltberg et al., 2019). To mimic irregularly sampled data, we follow the setup of Cheng et al. (2023) and randomly drop some observations with missing probability $p$. We also

simulate another dataset from a biological model, which describes metabolic iterations that break down glucose in cells. This is called *Glycolysis*, consisting of an SDE with 7 variables. As a preprocessing step, we standardised this dataset to avoid large differences in variable scales. Both datasets consist of $N = 10$ time series with sequence length $I = 100$ (before random drops), and have dimensionality 10 and 7, respectively. Note that we choose a large data sampling interval, as we want to test settings where observations are fairly sparse and the difficulty of correctly modelling continuous-time dynamics increases. The above data setup is different from Bellot et al. (2022); Cheng et al. (2023) where they use a single series with $I = 1000$ observations, which is more informative compared to our sparse setting. Refer to appendix D.3 and appendix D.4 for details.

The left two columns in table 1 compare the AUROC of *SCOTCH* to baselines for Lorenz. We can see that *SCOTCH* can effectively handle the irregularly sampled data compared to other baselines. Compared to NGM and CUTS, we can achieve much better results with small missingness and performs competitively with larger missingness. Rhino, VARLiNGaM and PCMCI+ perform poorly in comparison as they assume regularly sampled observations and are discrete in nature.

From the right column in table 1, *SCOTCH* outperforms the baselines by a large margin on Glycolysis. In particular, compared to the ODE-based NGM, *SCOTCH* clearly demonstrates the advantage of the proposed SDE framework in multiple time series settings. As we may have anticipated from the discussion in section 3.2, NGM can produce an incorrect model when multiple time series are sampled from a given SDE system. Another interesting observation is that *SCOTCH* is more robust when encountering data with different scales compared to NGM (refer to appendix D.4.3). This robustness is due to the stochastic nature of SDE compared to the deterministic ODE, where ODE can easily overshoot with less stable training behaviour. We can also see that *SCOTCH* has a significant advantage over both CUTS and Rhino, which do not model continuous-time dynamics.

| | Lorenz-96 | | Glycolysis |
|---|---|---|---|
| | $p = 0.3$ | $p = 0.6$ | Full |
| VARLiNGaM | 0.5102±0.025 | 0.4876±0.032 | 0.5082±0.009 |
| PCMCI+ | 0.4990±0.008 | 0.4952±0.021 | 0.4607±0.031 |
| NGM | 0.6788±0.009 | 0.6329±0.008 | 0.5953±0.018 |
| CUTS | 0.6042±0.015 | 0.6418±0.012 | 0.580±0.007 |
| Rhino | 0.5714±0.026 | 0.5123±0.025 | 0.520±0.015 |
| *SCOTCH* (ours) | **0.7279±0.017** | **0.6453±0.014** | **0.7113±0.012** |

Table 1: AUROC of synthetic datasets from *SCOTCH* and baselines. $p$ represents missing probability, and *Full* means complete data without missingness. Each number is the average over 5 runs.

## 6.2 DREAM3

We also evaluate *SCOTCH* performance on the DREAM3 datasets (Prill et al., 2010; Marbach et al., 2009), which have been adopted for assessing the performance of structure learning (Tank et al., 2021; Pamfil et al., 2020; Gong et al., 2022). These datasets contain *in silico* measurement of gene expression levels for 5 different structures. Each dataset corresponds to a particular gene expression network, and contains $N = 46$ time series of 100 dimensional variables, with $I = 21$ per series. The goal is to infer the underlying structures from each dataset. Following the same setup as (Gong et al., 2022; Khanna & Tan, 2019), we ignore all the self-connections by setting the edge probability to 0, and use AUROC as the performance metric. Appendix D.5 details the experiment setup, selected hyperparameters, and additional plots. We do not include VARLiNGaM since it cannot support time series where the dimensionality (100) is greater than the length (21). Also, due to the time series length, we decide not to test with irregularly sampled data. We use the reported numbers for Rhino and PCMCI+ in Gong et al. (2022) as the experimental setup is identical. For CUTS, we failed to reproduce the reported number in their paper, but we cite it for a fair comparison.

Table 2 shows the AUROC performances of *SCOTCH* and baselines. We can clearly observe that *SCOTCH* outperforms the other baselines with a large margin. This indicates the advantage of the SDE formulation compared to ODEs and discretized temporal models, even when we have complete and regularly sampled data. A more interesting observation is to compare Rhino with *SCOTCH*. As discussed before, as *SCOTCH* is the continuous version of Rhino, the advantage comes from the continuous formulation and the corresponding training objective eq. (15).

| | EColi1 | Ecoli2 | Yeast1 | Yeast2 | Yeast3 | Mean |
|---|---|---|---|---|---|---|
| PCMCI+ | 0.530±0.002 | 0.519±0.002 | 0.530 ±0.003 | 0.510±0.001 | 0.512 ± 0 | 0.520±0.004 |
| NGM | 0.611±0.002 | 0.595±0.005 | 0.597±0.006 | 0.563±0.006 | 0.535±0.004 | 0.580±0.007 |
| CUTS | 0.543±0.003 | 0.555±0.005 | 0.545±0.003 | 0.518±0.007 | 0.511±0.002 | 0.534±0.008 (0.591) |
| Rhino | 0.685±0.003 | 0.680±0.007 | 0.664±0.006 | 0.585±0.004 | 0.567±0.003 | 0.636±0.022 |
| *SCOTCH* (ours) | **0.752±0.008** | **0.705±0.003** | **0.712±0.003** | **0.622 ± 0.004** | **0.594± 0.001** | **0.677± 0.026** |

Table 2: AUROC for *SCOTCH* on DREAM3 100-dimensional datasets. Results are obtained by averaging over 5 runs. We cite the reported CUTS performance in parentheses.

| | Full | $p = 0.1$ | $p = 0.2$ |
|---|---|---|---|
| VARLiNGaM | 0.84±0 | 0.723±0.001 | 0.719±0.003 |
| PCMCI | 0.83±0 | 0.81±0.001 | 0.79±0.006 |
| NGM | 0.89 ± 0.009 | 0.86 ± 0.009 | 0.85 ±0.007 |
| CUTS | 0.89 ± 0.010 | 0.87 ± 0.008 | 0.87 ±0.011 |
| Rhino+NoInst | **0.95 ±0.001** | **0.93± 0.005** | **0.90±0.012** |
| *SCOTCH* (ours) | **0.95± 0.006** | 0.91±0.007 | 0.89±0.007 |
| Rhino | **0.99±0.001** | **0.98±0.004** | **0.97±0.003** |

Table 3: AUROC on Netsim dataset (subjects 2-6). Results are obtained by averaging over 5 runs.

## 6.3 NETSIM

Netsim consists of *blood oxygenation level dependent* imaging data. Following the same setup as Gong et al. (2022), we use subjects 2-6 to form the dataset, which consists of 5 time series. Each contains 15 dimensional observations with $I = 200$. The goal is to infer the underlying connectivity between different brain regions. Unlike Dream3, we include the self-connection edge for all methods. To evaluate the performance under irregularly sampled data, we follow the same setup as in the Lorenz and Cheng et al. (2023) to randomly drop observations with missing probability. Since it is very important to model instantaneous effects in Netsim (Gong et al., 2022), which *SCOTCH* cannot handle, we replace Rhino with Rhino+NoInst and PCMCI+ with PCMCI for fair comparison.

Table 3 shows the performance comparisons. We observe that *SCOTCH* significantly outperforms the other baselines and performs on par with Rhino+NoInst, which demonstrates its robustness towards smaller datasets and balance between true and false positive rates. Again, this confirms the modelling power of our approach compared to NGM and other baselines. Interestingly, Rhino-based approaches perform particularly well on the Netsim dataset. We suspect that the underlying generation mechanism can be better modelled with a discretized as opposed to continuous system.

## 7 CONCLUSION

We propose *SCOTCH*, a flexible continuous-time temporal structure learning method based on latent Itô diffusion. We leverage the variational inference framework to infer the posterior over latent states and the graph. Theoretically, we validate our approach by proving the structural identifiability of the Itô diffusion and latent formulation, and the consistency of the proposed variational framework. Empirically, we extensively evaluated our approach using synthetic and semi-synthetic datasets, where *SCOTCH* outperforms the baselines in both regularly and irregularly sampled data. There are three limitations that require further investigation. The first one is the inability to handle instantaneous effects, which can arise due to data aggregation. Another computational drawback is it scales linearly with the series length. This could be potentially fixed by incorporating an encoder network to infer latent states at arbitrary time points. Last but not least, the current formulation of *SCOTCH* cannot handle non-stationary systems due to the homogeneous drift and diffusion function. However, direct incorporation of time embeddings may break the theoretical guarantees without additional assumptions. Therefore, new theories and methodologies may be needed to tackle such a scenario. We leave these challenges for future work.

ACKNOWLEDGEMENTS

We thank the members of the Causica team at Microsoft Research for helpful discussions. We thank Colleen Tyler, Maria Defante, and Lisa Parks for conversations on real-world use cases that inspired this work. This work was done in part while Benjie Wang was visiting the Simons Institute for the Theory of Computing.

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
