## A  IDENTIFIABILITY OF STOCHASTIC DIFFERENTIAL EQUATIONS

### A.1  DEFINITIONS AND ASSUMPTIONS

In this part, we will introduce some basic definitions and assumptions required for the theory.

First, let us restate assumption 1.

**Assumption 1** (Global Lipschitz). *We assume that the drift and diffusion functions in eq. (5) satisfy the global Lipschitz constraints. Namely, we have*

$$|\boldsymbol{f}_\theta(\boldsymbol{x}) - \boldsymbol{f}_\theta(\boldsymbol{y})| + |\boldsymbol{g}_\theta(\boldsymbol{x}) - \boldsymbol{g}_\theta(\boldsymbol{y})| \leq C|\boldsymbol{x} - \boldsymbol{y}| \tag{6}$$

*for some constant $C$, $\boldsymbol{x}, \boldsymbol{y} \in \mathbb{R}^D$ and $|\cdot|$ is the corresponding $L_2$ norm for vector-valued functions and matrix norm for matrix-valued functions.*

This assumption regularizes the Itô diffusion to have a unique strong solution $\boldsymbol{X}_t$ to eq. (3), which is a standard assumption in the SDE literature. In addition, this diffusion satisfies the Feller continuous property, and its solution is a Feller process (Lemma 8.1.4 in Øksendal & Øksendal (2003)).

**Definition 1** (Feller process and semi-group). *A continuous time-homogeneous Markov family $\boldsymbol{X}_t$ is a Feller process when, for all $\boldsymbol{x} \in \mathbb{R}^D$, we have $\forall t, \boldsymbol{y} \to \boldsymbol{x} \Rightarrow \boldsymbol{X}_{y,t} \overset{d}{\to} \boldsymbol{X}_{x,t}$ and $t \to 0 \Rightarrow \boldsymbol{X}_{x,t} \overset{p}{\to} \boldsymbol{x}$ where $\overset{d}{\to}, \overset{p}{\to}$ means convergence in distribution and in probability, respectively, and $\boldsymbol{X}_{y,t}$ means the solution with $y$ as the initial condition. A semigroup of linear, positive, conservative contraction operators $\mathrm{T}_t$ is a Feller semigroup if, for every $\boldsymbol{f} \in C_0, \boldsymbol{x} \in \mathbb{R}^D$, we have $\mathrm{T}_t \boldsymbol{f} \in C_0$ and $\lim_{t \to 0} \mathrm{T}_t \boldsymbol{f}(\boldsymbol{x}) = \boldsymbol{f}(\boldsymbol{x})$, where $C_0$ is the space of continuous functions vanishing at infinity.*

Basically, the transition operator of a Feller process is a Feller semigroup. The reason we care about the Feller process is its nice properties related to its infinitesimal generators. In a nutshell, the distributional properties of the Feller process can be uniquely characterised by its generators.

**Definition 2** (Infinitesimal generator). *For a Feller process $\boldsymbol{X}_t$ with a Feller semigroup $\mathrm{T}_t$, we define the generator $\mathrm{A}$ by*

$$\mathrm{A}\, f = \lim_{t \downarrow 0} \frac{\mathrm{T}_t f - f}{t} \quad \text{for any } f \in D(A) \tag{17}$$

*where $D(A)$ is the domain of the generator, defined as the function space where the above limit exists.*

Next, let us restate assumption 2.

**Assumption 2** (Diagonal diffusion). *We assume that the diffusion function $\boldsymbol{g}_\theta$ outputs a non-zero diagonal matrix. That is, it can be simplified to a vector-valued function $\boldsymbol{g}_\theta(\boldsymbol{X}_t) : \mathbb{R}^D \to \mathbb{R}^D$.*

This is a key assumption for structure identifiability. For a general matrix diffusion function, it is easy to come up with unidentifiable examples (see Example 5.5 in (Hansen & Sokol, 2014)). For example, in a driftless process, the distribution of $\boldsymbol{X}_t$ will depend on $\boldsymbol{g}_G \boldsymbol{g}_G{}^T$, where it can have multiple factorizations that correspond to different graphs.

### A.2  STRUCTURE IDENTIFIABILITY FOR OBSERVATIONAL PROCESS

Now, let us re-state theorem 4.1:

**Theorem 4.1** (Structure identifiability of the observational process). *Given eq. (16), let us define another process with $\bar{\boldsymbol{X}}_t, \boldsymbol{G} \neq \bar{\boldsymbol{G}}, \bar{\boldsymbol{f}}_{\bar{G}}, \bar{\boldsymbol{g}}_{\bar{G}}$ and $\bar{\boldsymbol{W}}_t$. Then, under Assumptions 1-2, and with the same initial condition $\boldsymbol{X}(0) = \bar{\boldsymbol{X}}(0) = \boldsymbol{x}_0$, the solutions $\boldsymbol{X}_t$ and $\bar{\boldsymbol{X}}_t$ will have different distributions.*

To prove this theorem, we begin by establishing the corresponding result for the discretized Euler SEMs, and then build the connection to the Itô diffusion through the infinitesimal generator.

**Lemma A.1** (Identifiability of Euler SEM). *Assuming assumption 2 is satisfied with nonzero diagonal diffusion functions. For a Euler SEM defined as*

$$\boldsymbol{X}_{t+1}^\Delta = \boldsymbol{X}_t^\Delta + \boldsymbol{f}_G(\boldsymbol{X}_t^\Delta)\Delta + \boldsymbol{g}_G(\boldsymbol{X}_t^\Delta)\boldsymbol{\eta}_t, \quad \boldsymbol{\eta}_t \sim \mathcal{N}(0, \Delta\boldsymbol{I}), \tag{18}$$

*if we have another Euler SEM defined as*

$$\bar{\boldsymbol{X}}_{t+1}^{\Delta} = \bar{\boldsymbol{X}}_t^{\Delta} + \bar{\boldsymbol{f}}_{\bar{G}}(\bar{\boldsymbol{X}}_t^{\Delta})\Delta + \bar{\boldsymbol{g}}_{\bar{G}}(\bar{\boldsymbol{X}}_t^{\Delta})\bar{\boldsymbol{\eta}}_t, \quad \bar{\boldsymbol{\eta}}_t \sim \mathcal{N}(0, \Delta\boldsymbol{I}). \tag{19}$$

*Then their corresponding transition density $p(\boldsymbol{X}_{t+1}^{\Delta}|\boldsymbol{X}_t^{\Delta} = \boldsymbol{a}) = \bar{p}(\bar{\boldsymbol{X}}_{t+1}^{\Delta}|\bar{\boldsymbol{X}}_t^{\Delta} = \boldsymbol{a})$ for all $\boldsymbol{a} \in \mathbb{R}^D$ iff. $\boldsymbol{G} = \bar{\boldsymbol{G}}$, $\boldsymbol{f}_G = \bar{\boldsymbol{f}}_G$ and $|\boldsymbol{g}_G| = |\bar{\boldsymbol{g}}_G|$.*

*Proof.* If we have $\boldsymbol{G} = \bar{\boldsymbol{G}}$, $\boldsymbol{f} = \bar{\boldsymbol{f}}$ and $|\boldsymbol{g}| = |\bar{\boldsymbol{g}}|$, then it is trivial that their transition densities are the same since they define the same Euler SEM update equations (up to the sign of the diffusion term) with given initial conditions.

On the other hand, we know

$$p(\boldsymbol{X}_{t+1}^{\Delta}|\boldsymbol{X}_t^{\Delta} = \boldsymbol{a}) = \mathcal{N}(\boldsymbol{f}_G(\boldsymbol{a})\Delta + \boldsymbol{a}, \boldsymbol{g}_G{}^2(\boldsymbol{a})\Delta)$$
$$\bar{p}(\bar{\boldsymbol{X}}_{t+1}^{\Delta}|\bar{\boldsymbol{X}}_t^{\Delta} = \boldsymbol{a}) = \mathcal{N}(\bar{\boldsymbol{f}}_{\bar{G}}(\boldsymbol{a})\Delta + \boldsymbol{a}, \bar{\boldsymbol{g}}_{\bar{G}}^2(\boldsymbol{a})\Delta)$$

Thus, if two conditional distributions match, we have

$$\boldsymbol{f}_G(\boldsymbol{a})\Delta = \bar{\boldsymbol{f}}_{\bar{G}}(\boldsymbol{a})\Delta \quad \boldsymbol{g}_G{}^2(\boldsymbol{a})\Delta = \bar{\boldsymbol{g}}_{\bar{G}}^2(\boldsymbol{a})\Delta \tag{20}$$

Since $\Delta > 0$, we have $\boldsymbol{f}_G(\boldsymbol{a}) = \bar{\boldsymbol{f}}_{\bar{G}}(\boldsymbol{a})$, $\boldsymbol{g}_G{}^2(\boldsymbol{a}) = \bar{\boldsymbol{g}}_{\bar{G}}^2(\boldsymbol{a})$ for all $\boldsymbol{a} \in \mathbb{R}^D$. From the diagonal diffusion assumption, we know $|\boldsymbol{g}_G(\boldsymbol{a})| = |\bar{\boldsymbol{g}}_{\bar{G}}(\boldsymbol{a})|$.

Now, assume for contradiction that $\boldsymbol{G} \neq \bar{\boldsymbol{G}}$; then there exists $X_{t,i}^{\Delta} \to X_{t+1,j}^{\Delta}$ in $\boldsymbol{G}$ but not in $\bar{\boldsymbol{G}}$. Then we have by definition that $\frac{\partial \bar{f}_j(\boldsymbol{X}_t^{\Delta}, \bar{\boldsymbol{G}})}{\partial X_{t,i}^{\Delta}} = 0$ and $\frac{\partial \bar{g}_j(\boldsymbol{X}_t^{\Delta}, \bar{\boldsymbol{G}})}{\partial X_{t,i}^{\Delta}} = 0$ for all $\boldsymbol{X}_t^{\Delta}$, and also $\frac{\partial f_j(\boldsymbol{X}_t^{\Delta}, \boldsymbol{G})}{\partial X_{t,i}^{\Delta}} \neq 0$ or $\frac{\partial g_j(\boldsymbol{X}_t^{\Delta}, \boldsymbol{G})}{\partial X_{t,i}^{\Delta}} \neq 0$ for some $\boldsymbol{X}_t^{\Delta}$. In the former case, if $\frac{\partial f_j(\boldsymbol{X}_t^{\Delta}, \boldsymbol{G})}{\partial X_{t,i}^{\Delta}} \neq 0$ but $\frac{\partial \bar{f}_j(\boldsymbol{X}_t^{\Delta}, \bar{\boldsymbol{G}})}{\partial X_{t,i}^{\Delta}} = 0$ for some $\boldsymbol{X}_t^{\Delta}$, we have a contradiction to $\boldsymbol{f}_G(\boldsymbol{a}) = \bar{\boldsymbol{f}}_{\bar{G}}(\boldsymbol{a})$ for $\boldsymbol{a} \in \mathbb{R}^D$. A similar analysis can be done in the latter case for $\boldsymbol{g}_G, \bar{\boldsymbol{g}}_{\bar{G}}$. Thus, we have $\boldsymbol{G} = \bar{\boldsymbol{G}}$, $\boldsymbol{f}_G = \bar{\boldsymbol{f}}_G$ and $|\boldsymbol{g}_G| = |\bar{\boldsymbol{g}}_G|$. $\qquad\square$

Next, we will prove a lemma that builds a bridge between the generator of the Itô diffusion and its corresponding Euler SEM.

**Lemma A.2** (Generator characterises Euler SEM). *Assume that assumptions 1 and 2. For an Itô diffusion defined as eq. (16), we denote its corresponding variables in Euler SEM with $\Delta$ discretization as $\boldsymbol{X}^{\Delta}$. Similarly, if we have an alternative Itô diffusion defined with $\bar{\boldsymbol{f}}_{\bar{G}}, \bar{\boldsymbol{g}}_{\bar{G}}$ and $\bar{\boldsymbol{G}}$, ae define the corresponding Euler SEM variables $\bar{\boldsymbol{X}}^{\Delta}$. Then, the generators of the Itô diffusions $A = \bar{A}$ iff. their Euler SEM variables have the same distribution with given initial conditions.*

*Proof.* First, assume $A = \bar{A}$, then for any $h \in C_0^2$ (twice continuously differentiable functions vanishing at infinity), we can define the generator for Itô diffusion as

$$A h(\boldsymbol{x}) = \sum_d f_d(\boldsymbol{x}, \boldsymbol{G})\frac{\partial h(\boldsymbol{x})}{\partial x_d} + \frac{1}{2}\sum_d g_d^2(\boldsymbol{x}, \boldsymbol{G})\frac{\partial^2 h(\boldsymbol{x})}{\partial x_d^2} \tag{21}$$

Similarly, we can define $\bar{A}$. From Lemma A.3 (Hansen & Sokol, 2014), we know if $A = \bar{A}$, then $\boldsymbol{G} = \bar{\boldsymbol{G}}$, $\boldsymbol{f}(\cdot, \boldsymbol{G}) = \bar{\boldsymbol{f}}(\cdot, \bar{\boldsymbol{G}})$ and $\boldsymbol{g}^2(\cdot, \boldsymbol{G}) = \bar{\boldsymbol{g}}^2(\cdot, \bar{\boldsymbol{G}})$ for $\boldsymbol{x} \in \mathbb{R}^D$. Therefore, by the definition of Euler SEM (eq. (4)), it is trivial that they define the same transition density $p(\boldsymbol{X}_{t+1}^{\Delta}|\boldsymbol{X}_t^{\Delta} = \boldsymbol{a}) = \bar{p}(\bar{\boldsymbol{X}}_{t+1}^{\Delta}|\bar{\boldsymbol{X}}_t^{\Delta} = \boldsymbol{a})$ for $\boldsymbol{a} \in \mathbb{R}^D$.

On the other hand, if the two Euler SEMs define the same transition densities, then from Lemma A.1, we have $\boldsymbol{f}_G = \bar{\boldsymbol{f}}_G$, $|\boldsymbol{g}_G| = |\bar{\boldsymbol{g}}_G|$ and $\boldsymbol{G} = \bar{\boldsymbol{G}}$. Then from eq. (21), $A = \bar{A}$. $\qquad\square$

Finally, the following lemma shows why we care about the infinitesimal generator for the Feller process.

**Lemma A.3** (Generator uniquely determines Feller semigroup). *Let us define the Feller semigroup transition operator $T_t$ and $\bar{T}_t$ associated with generator $A$, $\bar{A}$. Then, $T_t = \bar{T}_t$ iff. $A = \bar{A}$.*

*Proof.* We define the resolvent of a Feller process with $\lambda > 0$ as:

$$R_\lambda f = \int_0^\infty \exp(-\lambda t) \mathrm{T}_t f \, dt \tag{22}$$

with $f \in C_0$. This is the Laplace transform of $\mathrm{T}_t f$. From Øksendal & Øksendal (2003), we know $R_\lambda = (\lambda \boldsymbol{I} - \mathrm{A})^{-1}$. Therefore, if $\mathrm{A} = \bar{\mathrm{A}}$, then for $\lambda > 0$, the resolvent $R_\lambda = (\lambda \boldsymbol{I} - \mathrm{A})^{-1} = (\lambda \boldsymbol{I} - \bar{\mathrm{A}})^{-1} = \bar{R}_\lambda$. Therefore, for all $h \in C_0$, they define the same Laplace transform of $\mathrm{T}_t h$. From the uniqueness of Laplace transform, we have $\mathrm{T}_t = \bar{\mathrm{T}}_t$.

Similarly, if $\mathrm{T}_t = \bar{\mathrm{T}}_t$, we have $R_\lambda = \bar{R}_\lambda$ from the definition of resolvent. Thus, $\mathrm{A} = \lambda \boldsymbol{I} - R_\lambda^{-1} = \lambda \boldsymbol{I} - \bar{R}_\lambda^{-1} = \bar{\mathrm{A}}$. □

Now, we can prove theorem 4.1.

*Proof.* Suppose we have two different observation process defined with $\boldsymbol{G} \neq \bar{\boldsymbol{G}}$. Then, from Lemma A.1, with any $\Delta > 0$, their Euler transition distribution $\bar{p}(\bar{\boldsymbol{X}}_{t+1}^\Delta | \bar{\boldsymbol{X}}_t^\Delta = \boldsymbol{a}) \neq p(\boldsymbol{X}_{t+1}^\Delta | \boldsymbol{X}_t^\Delta = \boldsymbol{a})$. Thus, from Lemma A.2, these two Itô diffusions have different generators $\mathrm{A} \neq \bar{\mathrm{A}}$. From assumption 1, the solutions of these two Itô diffusions are Feller processes. From Lemma A.3, if $\mathrm{A} \neq \bar{\mathrm{A}}$, their semigroup $\mathrm{T}_t \neq \bar{\mathrm{T}}_t$, which results in different observation distributions of $\boldsymbol{X}_t, \bar{\boldsymbol{X}}_t$. □

## A.3 IDENTIFIABILITY OF LATENT SDE

We begin by re-stating theorem 4.2:

**Theorem 4.2** (Structural identifiability with latent formulation)**.** *Consider the distributions $p, \bar{p}$ defined by the latent model in eq. (5) with $(\boldsymbol{G}, \boldsymbol{Z}, \boldsymbol{X}, \boldsymbol{f}_G, \boldsymbol{g}_G), (\bar{\boldsymbol{G}}, \bar{\boldsymbol{Z}}, \bar{\boldsymbol{X}}, \bar{\boldsymbol{f}}_{\bar{G}}, \bar{\boldsymbol{g}}_{\bar{G}})$ respectively, where $\boldsymbol{G} \neq \bar{\boldsymbol{G}}$. Further, let $t_1, \ldots, t_I$ be the observation times. Then, under Assumptions 1 and 2:*

1. *if $t_{i+1} - t_i = \Delta$ for all $i \in 1, \ldots, I-1$, then $p^\Delta(\boldsymbol{X}_{t_1}, \ldots, \boldsymbol{X}_{t_I}) \neq \bar{p}^\Delta(\bar{\boldsymbol{X}}_{t_1}, \ldots, \bar{\boldsymbol{X}}_{t_I})$, where $p^\Delta$ is the density generated by the Euler discretized eq. (9) for $\boldsymbol{Z}_t$;*

2. *if we have a fixed time range $[0, T]$, then the path probability $p(\boldsymbol{X}_{t_1}, \ldots, \boldsymbol{X}_{t_I}) \neq \bar{p}(\bar{\boldsymbol{X}}_{t_1}, \ldots, \bar{\boldsymbol{X}}_{t_I})$ under the limit of infinite data ($I \to \infty$).*

We follow the same proof strategy as Hasan et al. (2021); Khemakhem et al. (2020).

*Proof.* Let's assume $p(\boldsymbol{X}_{t_1}, \ldots, \boldsymbol{X}_{t_I}) = \bar{p}(\bar{\boldsymbol{X}}_{t_1}, \ldots, \bar{\boldsymbol{X}}_{t_I})$ even though $\boldsymbol{G} \neq \bar{\boldsymbol{G}}$. Then, for any $t_{i+1}$ and $t_i$, we have $p(\boldsymbol{X}_{t_{i+1}}, \boldsymbol{X}_{t_i}) = \bar{p}(\bar{\boldsymbol{X}}_{t_{i+1}}, \bar{\boldsymbol{X}}_{t_i})$. Then, we can write

$$p(\boldsymbol{X}_{t_{i+1}}, \boldsymbol{X}_{t_i}) = \int p(\boldsymbol{Z}_{t_{i+1}}, \boldsymbol{Z}_{t_i}, \boldsymbol{X}_{t_{i+1}}, \boldsymbol{X}_{t_i}) d\boldsymbol{Z}_{t_{i+1}} d\boldsymbol{Z}_{t_i}$$

$$= \int p_z(\boldsymbol{Z}_{t_{i+1}}, \boldsymbol{Z}_{t_i}) p_\epsilon(\boldsymbol{X}_{t_{i+1}} - \boldsymbol{Z}_{t_{i+1}}) p_\epsilon(\boldsymbol{X}_{t_i} - \boldsymbol{Z}_{t_i}) d\boldsymbol{Z}_{t_{i+1}} d\boldsymbol{Z}_{t_i}$$

$$= [(p_\epsilon \times p_\epsilon) * p_z](\boldsymbol{X}_{t_{i+1}}, \boldsymbol{X}_{t_i})$$

where $p_\epsilon$ is the noise density for the added observational noise $\boldsymbol{\epsilon}$, $p_z$ is the joint density defined by latent Itô diffusion and $*$ is the convolution operator. Thus, by applying the Fourier transform $\mathcal{F}$, we obtain

$$\mathcal{F}(p_\epsilon \times p_\epsilon)(\omega) \times \mathcal{F}(p_z)(\omega) = \mathcal{F}(p_\epsilon \times p_\epsilon)(\omega) \times \mathcal{F}(\bar{p}_z)(\omega) \tag{23}$$

So $\mathcal{F}(p_z) = \mathcal{F}(\bar{p}_z)$. Then, by inverse Fourier transform, we have $p_z(\boldsymbol{Z}_{t_{i+1}}, \boldsymbol{Z}_{t_i}) = \bar{p}_z(\bar{\boldsymbol{Z}}_{t_{i+1}}, \bar{\boldsymbol{Z}}_{t_i})$.

If the above distributions are obtained by discretizing the Itô diffusion with a fixed step size $\Delta$, they become the corresponding discretized distribution $p^\Delta(\boldsymbol{Z}_{t_{i+1}}^\Delta, \boldsymbol{Z}_{t_i}^\Delta)$ (i.e. defined by Euler SEM). Then the transition density $p^\Delta(\boldsymbol{Z}_{t_{i+1}}^\Delta | \boldsymbol{Z}_{t_i}^\Delta) = \bar{p}^\Delta(\bar{\boldsymbol{Z}}_{t_{i+1}}^\Delta | \bar{\boldsymbol{Z}}_{t_i}^\Delta)$. From Lemma A.1, we have $\boldsymbol{G} = \bar{\boldsymbol{G}}$, resulting in a contradiction. Thus, $p^\Delta(\boldsymbol{X}_{t_1}, \ldots, \boldsymbol{X}_{t_I}) \neq \bar{p}^\Delta(\bar{\boldsymbol{X}}_{t_1}, \ldots, \bar{\boldsymbol{X}}_{t_I})$.

If we have a fixed time range $[0, T]$, then, when we have infinite observations $I \to \infty$, the observation time $t$ follows an independent temporal point process with intensity $\lim_{dt \to 0} Pr(\text{observe in } [t, t+dt] | \mathcal{H}_t) > 0$ where $\mathcal{H}_t$ is the filtration. Thus, for arbitrary time interval $\Delta > 0$, we have $p(\boldsymbol{Z}_{t+\Delta}, \boldsymbol{Z}_t) = \bar{p}(\bar{\boldsymbol{Z}}_{t+\Delta}, \bar{\boldsymbol{Z}}_t)$. Since this holds for arbitrarily small $\Delta > 0$,

this equality in densities means they define the same transition density $p(\boldsymbol{Z}_{t+\Delta}|\boldsymbol{Z}_t) = \bar{p}(\bar{\boldsymbol{Z}}_{t+\Delta}|\bar{\boldsymbol{Z}}_t)$ as $\Delta \to 0$. By definition of the Feller transition semigroup, we have $T_t = \bar{T}_t$. From Lemma A.3, $A = \bar{A}$ and $\boldsymbol{G} = \bar{\boldsymbol{G}}$ (Lemma A.2, A.1). This leads to contradiction, meaning that $p(\boldsymbol{X}_{t_1}, \ldots, \boldsymbol{X}_{t_I}) \neq \bar{p}(\boldsymbol{X}_{t_1}, \ldots, \boldsymbol{X}_{t_I})$ when $I \to \infty$. $\qquad\square$

## A.4 RECOVERY OF THE GROUND TRUTH GRAPH

Before diving into the proof of theorem 4.3, we introduce some necessary assumptions:

**Assumption 3** (Correctly specified model). *We say a model is correctly specified w.r.t. the ground truth data generating mechanism iff. there exists a model parameter such that the model coincides with the generating mechanism.*

**Assumption 4** (Expressive posterior process). *For a given prior parameter $\theta$, we say the approximate posterior process (eq. (12)) is expressive enough if there exists a measurable function $\boldsymbol{u}(\boldsymbol{Z}_t)$ such that (i) $\boldsymbol{g}_G(\boldsymbol{Z}_t)\boldsymbol{u}(\boldsymbol{Z}_t) = \boldsymbol{f}_G(\boldsymbol{Z}_t) - \boldsymbol{h}_\phi(\boldsymbol{Z}, t, \boldsymbol{G})$; (ii) $\boldsymbol{u}(\boldsymbol{Z}_t)$ satisfies Novikov's condition and (iii) we define*

$$\boldsymbol{M}_T = \exp\left(-\frac{1}{2}\int_0^T |\boldsymbol{u}(\boldsymbol{Z}_t)|^2 dt - \int_0^T \boldsymbol{u}(\boldsymbol{Z}_t)^T d\boldsymbol{W}_t\right) \tag{24}$$

*and for given latent states $\boldsymbol{Z}_{t_1}, \ldots, \boldsymbol{Z}_{t_I}$ and corresponding observations $\boldsymbol{X}_{t_1}, \ldots, \boldsymbol{X}_{t_I}$ with $0 \leq t_1 \leq t_2 \leq \ldots \leq t_I \leq T$, $\boldsymbol{M}_T$ can approximate the following arbitrarily well:*

$$\boldsymbol{M}_T \approx \frac{\prod_{i=1}^I p(\boldsymbol{X}_{t_i}|\boldsymbol{Z}_{t_i}, \boldsymbol{G})}{p(\boldsymbol{X}_{t_1}, \ldots, \boldsymbol{Z}_{t_I}|\boldsymbol{G})} \tag{25}$$

This assumption is to make sure the approximate posterior process is expressive enough to make the variational bound tight. Since we use neural networks to define the drift and diffusion functions, the corresponding approximate posterior is flexible. In fact, Tzen & Raginsky (2019b) showed that the diffusion defined by eq. (12) can be used to obtain samples from any distributions whose Radon-Nikodym derivative w.r.t. standard Gaussian measure can be represented by neural networks. Due to the universal approximation theorem for neural networks (Hornik et al., 1989), the corresponding posterior is indeed flexible.

First, we can re-write the ELBO (eq. (15)) (for a single time series) as the following:

$$\log p(\boldsymbol{X}_{t_1}, \ldots, \boldsymbol{X}_{t_I}) \geq \mathbb{E}_{\boldsymbol{G} \sim q_\phi(\boldsymbol{G})}\left[\mathbb{E}_P\left[\sum_{i=1}^I \log p(\boldsymbol{X}_{t_i}|\tilde{\boldsymbol{Z}}_{t_i}) - \frac{1}{2}\int_0^T |\boldsymbol{u}(\tilde{\boldsymbol{Z}}_t)|^2 dt\right]\right] - D_{\mathrm{KL}}[q_\phi(\boldsymbol{G})\|p(\boldsymbol{G})] \tag{26}$$

where $P$ is the probability measure in the filtered probability space $(\Sigma, \mathcal{F}, \{\mathcal{F}\}_{0 \leq t \leq T}, P)$, and $\tilde{\boldsymbol{Z}}_t$ is the path sampled from the approximate posterior process (eq. (12)). Let's restate the theorem:

**Theorem 4.3** (Consistency of variational formulation). *Suppose Assumptions 1-4 are satisfied for the latent formulation (eq. (5)). Then, for a fixed observation time range $[0, T]$, as the number of observations $I \to \infty$, when ELBO (eq. (15)) is maximised, $q_\phi(\boldsymbol{G}) = \delta(\boldsymbol{G}^*)$, where $\boldsymbol{G}^*$ is the ground truth graph, and the latent formulation recovers the underlying ground truth mechanism.*

*Proof.* First, we want to show that the term inside the $\mathbb{E}_{\boldsymbol{G} \sim q_\phi(\boldsymbol{G})}[\cdot]$ represents the $\log p(\boldsymbol{X}_{t_1}, \ldots, \boldsymbol{X}_{t_I}|\boldsymbol{G})$.

We define a measurable function $\boldsymbol{u}(\boldsymbol{Z}_t)$ that satisfies Novikov's condition. From the Girsanov theorem, we can construct another process

$$d\tilde{\boldsymbol{W}} = \boldsymbol{u}(\boldsymbol{Z}_t)dt + d\boldsymbol{W}_t \tag{27}$$

and another probability measure $Q$ s.t. $\tilde{\boldsymbol{W}}$ is a Brownian motion under measure $Q$ with

$$\frac{dQ}{dP} = \exp\left(-\frac{1}{2}\int_0^T |\boldsymbol{u}(\boldsymbol{Z}_t)|^2 dt - \int_0^T \boldsymbol{u}(\boldsymbol{Z}_t)^T d\boldsymbol{W}_t\right) \tag{28}$$

where $P$ is the probability measure associated with the original Brownian motion $\boldsymbol{W}_t$. From Boué & Dupuis (1998); Tzen & Raginsky (2019a), we have the following variational formulation:

$$\log \mathbb{E}_P \left[ \prod_{i=1}^{I} p(\boldsymbol{X}_{t_i} | \boldsymbol{Z}_{t_i}, \boldsymbol{G}) \right] = \sup_{Q \in \mathbb{P}} \left\{ -D_{\mathrm{KL}}[Q\|P] + \mathbb{E}_Q \left[ \sum_{i=1}^{I} \log p(\boldsymbol{X}_{t_i} | \boldsymbol{Z}_{t_i}, \boldsymbol{G}) \right] \right\} \quad (29)$$

where $\mathbb{P}$ represents the set of probability measures for the path $\boldsymbol{Z}_t$. Assume measure $Q$ is constructed by $\boldsymbol{u}$, we can write down $D_{\mathrm{KL}}[Q\|P]$ by substituting eq. (28):

$$\begin{aligned}
D_{\mathrm{KL}}[Q\|P] &= \mathbb{E}_Q[\log \frac{dQ}{dP}] \\
&= \int \left[ -\frac{1}{2} \int_0^T |\boldsymbol{u}(\boldsymbol{Z}_t)|^2 dt - \int_0^T \boldsymbol{u}(\boldsymbol{Z}_t)^T d\boldsymbol{W}_t \right] dQ \\
&= \int \left[ -\frac{1}{2} \int_0^T |\boldsymbol{u}(\boldsymbol{Z}_t)|^2 dt + \int_0^T |\boldsymbol{u}(\boldsymbol{Z}_t)|^2 dt \right] dQ \\
&= \mathbb{E}_Q \left[ \frac{1}{2} \int_0^T |\boldsymbol{u}(\boldsymbol{Z}_t)|^2 dt \right].
\end{aligned}$$

The third equality can be obtained by manipulating eq. (27):

$$\boldsymbol{u}(\boldsymbol{Z}_t)^T d\tilde{\boldsymbol{W}}_t = |\boldsymbol{u}(\boldsymbol{Z}_t)|^2 dt + \boldsymbol{u}(\boldsymbol{Z}_t)^T d\boldsymbol{W}_t$$

$$\Rightarrow \underbrace{\mathbb{E}_Q \left[ \int_0^T \boldsymbol{u}^T d\tilde{\boldsymbol{W}}_t \right]}_{=0} = \mathbb{E}_Q \left[ \int_0^T |\boldsymbol{u}|^2 dt \right] + \mathbb{E}_Q \left[ \int_0^T \boldsymbol{u}^T d\boldsymbol{W}_t \right]$$

The highlighted term is 0 due to the martingale property under measure $Q$. Thus, we have

$$\mathbb{E}_Q \left[ \int_0^T \boldsymbol{u}^T d\boldsymbol{W}_t \right] = -\mathbb{E}_Q \left[ \int_0^T |\boldsymbol{u}|^2 dt \right] \quad (30)$$

Now, let's define

$$\boldsymbol{u}(\boldsymbol{Z}_t) = \boldsymbol{g}_G(\boldsymbol{Z}_t)^{-1} [\boldsymbol{f}_\theta(\boldsymbol{Z}_t, \boldsymbol{G}) - \boldsymbol{h}_\phi(\boldsymbol{Z}_t, t, \boldsymbol{G})] \quad (31)$$

Note that this is different to the original $\boldsymbol{u}$ (eq. (14)) by a minus sign. But this does not affect the derivation because we care about $\boldsymbol{u}^2$. By simple manipulation of eq. (27), we have

$$\boldsymbol{h}_\phi(\boldsymbol{Z}_t, t, \boldsymbol{G})dt + \boldsymbol{g}_G(\boldsymbol{Z}_t)d\tilde{\boldsymbol{W}}_t = \boldsymbol{f}_G(\boldsymbol{Z}_t)dt + \boldsymbol{g}_G(\boldsymbol{Z}_t)d\boldsymbol{W}_t \quad (32)$$

This means the prior process (eq. (9)) under probability measure $Q$ is equivalent to the posterior process (eq. (12)) under probability measure $P$. Next, we can change the probability measure of eq. (29):

$$\begin{aligned}
&\sup_{Q \in \mathbb{P}} \left\{ \mathbb{E}_Q \left[ \sum_{i=1}^{I} \log p(\boldsymbol{X}_{t_i} | \boldsymbol{Z}_{t_i}, \boldsymbol{G}) - \frac{1}{2} \int_0^T |\boldsymbol{u}(\boldsymbol{Z}_t)|^2 dt \right] \right\} \\
&= \sup_{\boldsymbol{u}} \left\{ \mathbb{E}_P \left[ \sum_{i=1}^{I} \log p(\boldsymbol{X}_{t_i} | \tilde{\boldsymbol{Z}}_{t_i}, \boldsymbol{G}) - \frac{1}{2} \int_0^T |\boldsymbol{u}(\tilde{\boldsymbol{Z}}_t)|^2 dt \right] \right\}
\end{aligned}$$

where the second equality is obtained since $\frac{dQ}{dP}$ is fully determined by function $\boldsymbol{u}$, and $\tilde{\boldsymbol{Z}}_t$ is obtained from the posterior process eq. (12). This equation is exactly the term inside $\mathbb{E}_{\boldsymbol{G} \sim q(\boldsymbol{G})}[\cdot]$ since $\boldsymbol{g}_G(\boldsymbol{Z}_t)^{-2}[\boldsymbol{f}_G(\boldsymbol{Z}_t) - \boldsymbol{h}_\phi(\boldsymbol{Z}_t, t, \boldsymbol{G})]^2 = \boldsymbol{g}_G(\boldsymbol{Z}_t)^{-2}[\boldsymbol{h}_\phi(\boldsymbol{Z}_t, t, \boldsymbol{G}) - \boldsymbol{f}_G(\boldsymbol{Z}_t)]^2$.

From Proposition 2.4.2 in (Dupuis & Ellis, 2011), the supremum is uniquely obtained at

$$\frac{dQ*}{dP} = \frac{\prod_{i=1}^{I} p(\boldsymbol{X}_{t_i} | \boldsymbol{Z}_{t_i}, \boldsymbol{G})}{\mathbb{E}_P[\prod_{i=1}^{I} p(\boldsymbol{X}_{t_i} | \boldsymbol{Z}_{t_i}, \boldsymbol{G})]}.$$

From assumption 4, the measure $Q$ induced by $\boldsymbol{u}$ can approximate the above arbitrarily well. Thus, the eq. (26) can be written as:

$$\sup_{q(\boldsymbol{G}),\theta,\phi} \text{ELBO} = \sup_{q(\boldsymbol{G})} \left[ \log p(\boldsymbol{X}_{t_1},\ldots,\boldsymbol{X}_{t_I}|\boldsymbol{G}) \right] - D_{\text{KL}}[q(\boldsymbol{G})\|p(\boldsymbol{G})]$$

We divide the ELBO by $\frac{1}{I}$, and let $I \to \infty$, we have

$$\lim_{I\to\infty} \frac{1}{I} \left[ \log p(\boldsymbol{X}_{t_1},\ldots,\boldsymbol{X}_{t_I}|\boldsymbol{G}) \right] - \frac{1}{I} KL[q(\boldsymbol{G})\|p(\boldsymbol{G})]$$

$$= \lim_{I\to\infty} \frac{1}{I} \left[ \log p(\boldsymbol{X}_{t_1},\ldots,\boldsymbol{X}_{t_I}|\boldsymbol{G}) \right]$$

$$\leq \lim_{I\to\infty} \frac{1}{I} \log p(\boldsymbol{X}_{t_1},\ldots,\boldsymbol{X}_{t_I};\boldsymbol{G}^*)$$

where the first equality is obtained by the fact $D_{\text{KL}}[q(\boldsymbol{G})\|p(\boldsymbol{G})] < \infty$, and the second inequality is due to the property of the ground truth likelihood. From the identifiability theorem 4.2, the equality is uniquely obtained at $q(\boldsymbol{G}) = \delta(\boldsymbol{G}^*)$, and the learned system recovers the true generating mechanism under infinite data limits. $\qquad\square$

# B   MODEL ARCHITECTURE

In this section, we describe the model architecture details used in our experiments for *SCOTCH*.

**Prior Drift Function and Diffusion Function**   As described in Section 3, following Geffner et al. (2022), we use the following design for the prior drift function $\boldsymbol{f}_{G,d}(\boldsymbol{Z}_t)$ and diffusion function $\boldsymbol{g}_{G,d}(\boldsymbol{Z}_t)$:

$$\boldsymbol{f}_{G,d}(\boldsymbol{Z}_t) = \zeta\left( \sum_{i=1}^{D} G_{i,d} l(Z_{t,i}, \boldsymbol{e}_i), \boldsymbol{e}_d \right) \tag{33}$$

where $\zeta : \mathbb{R}^{D_g \times D_e} \to \mathbb{R}^D$, $l : \mathbb{R}^{D \times D_e} \to \mathbb{R}^{D_g}$ are neural networks, and $\boldsymbol{e}_i \in \mathbb{R}^{D_e}$ is a trainable node embedding for the $i^{\text{th}}$ node. The use of node embeddings means that we only need to train two neural networks, regardless of the latent dimensionality $D$.

We implement both the prior drift and diffusion function using $D_e = D_g = 32$, and as neural networks with two hidden layers of size $\max(2*D, D_e)$ with residual connections.

**Posterior Drift Function**   In Section 3.1, we described the posterior SDE $d\tilde{\boldsymbol{Z}}_t^{(n)} = \boldsymbol{h}_\psi(\tilde{\boldsymbol{Z}}_t^{(n)}, t; \boldsymbol{G}, \boldsymbol{X}^{(n)})dt + \boldsymbol{g}_G(\tilde{\boldsymbol{Z}}_t^{(n)})d\boldsymbol{W}_t$, with posterior drift function $\boldsymbol{h}_\psi(\tilde{\boldsymbol{Z}}_t^{(n)}, t; \boldsymbol{G}, \boldsymbol{X}^{(n)})$. We now elaborate on how this is implemented.

We design an encoder $\boldsymbol{K}_\psi(t, \boldsymbol{G}, \boldsymbol{X})$, that takes as input the time $t$, a graph $\boldsymbol{G}$ and time series $\boldsymbol{X} = \{\boldsymbol{X}_{t_1}, ...\boldsymbol{X}_{t_I}\}$, and outputs a *context vector* $\boldsymbol{c} \in \mathbb{R}^{D_c}$. This encoder consists of a GRU (Cho et al., 2014) that takes as input all future observations (i.e. $\boldsymbol{X}_{t_i}$ s.t. $t_i > t$) in reverse order; and a single linear layer which takes the input (i) the hidden state of the GRU, and (ii) the flattend graph matrix $\boldsymbol{G}$, and output the context vector $\boldsymbol{c}$. Note that the GRU only takes as input future observations as the future evolution of the latent state is conditionally independent of past observations given the current latent state. We implement the GRU with hidden size 128, and choose $D_c = 64$ for the size of the context vector.

Then, the posterior drift function $\boldsymbol{h}_\psi(\tilde{\boldsymbol{Z}}_t^{(n)}, t; \boldsymbol{G}, \boldsymbol{X}^{(n)})$ is implemented as a neural network that takes as input $\tilde{\boldsymbol{Z}}_t^{(n)}$ and the context vector $\boldsymbol{c}$ computed by the encoder, and outputs a vector of dimension $D$. This neural network is a MLP with 1 hidden layer of size 128.

**Posterior Mean and Covariance**   In Section 3.1, we also have posterior mean and covariance functions $\boldsymbol{\mu}_\psi(\boldsymbol{G}, \boldsymbol{X}^{(n)}) : \{0,1\}^{D \times D} \times \mathbb{R}^D \to \mathbb{R}^D$ and $\boldsymbol{\Sigma}_\psi(\boldsymbol{G}, \boldsymbol{X}^{(n)}) : \{0,1\}^{D \times D} \times \mathbb{R}^D \to \mathbb{R}^{D \times D}$ for the initial state. We reuse the encoder $\boldsymbol{K}_\psi(t, \boldsymbol{G}, \boldsymbol{X})$ with $t = 0$ to encode the entire time series and graph, and then implement $\boldsymbol{\mu}_\psi, \boldsymbol{\Sigma}_\psi$ as a linear transformation of the context vector (i.e. a single linear layer).

**Posterior Graph Distribution** In Section 3.1, we introduced a variational approximation $q_\phi(\boldsymbol{G})$ to the true posterior $p(\boldsymbol{G}|\boldsymbol{X}^{(1)}, ..., \boldsymbol{X}^{(N)})$. To implement this, we use a product of independent Bernoulli distributions for each edge. That is, we have:

$$q_\phi(\boldsymbol{G}) = \prod_{i,j} \phi_{ij}^{G_{i,j}}(1 - \phi_{ij})^{(1-G_{i,j})} \tag{34}$$

where $\phi_{ij} \in [0, 1]$ are learnable parameters corresponding to the probability of edge $i \to j$ being present.

**Observational Likelihood** We choose the observational noise $p_{\epsilon_d}$ in the model to follow a standard Laplace distribution with location $\mu = 0$ and scale $b = 0.01$.

## C    BASELINES

We use the following baselines for all our experiments to evaluate the performance of *SCOTCH*.

- PCMCI+:Runge (2018; 2020) proposed a constraint-based causal discovery methods for time series, which leverage the momentary conditional independence test to simultaneously detect the lagged parents and instantaneous effects. This is an improvement over its predecessor called PCMCI, which cannot handle instantaneous effects. In our experiments, we use PCMCI for Netsim and PCMCI+ for the other datasets. We use the opensourced implementation *Tigramite* (`https://github.com/jakobrunge/tigramite`).

- VARLiNGaM: Hyvärinen et al. (2010) proposed a linear vector auto-regressive model to learn from time series observations. It is an extension of LiNGaM (Shimizu et al., 2006), where its structural identifiability is guaranteed through the non-Gaussian noise assumption. The major limitation is its linear and discrete nature, which cannot model complex interactions and continuous systems. We also use the opensourced *LiNGaM* package (`https://lingam.readthedocs.io/en/latest/tutorial/var.html`)

- CUTS: CUTS (Cheng et al., 2023) is based on Granger causality, and designed for inferring structures from irregularly sampled time series. It treats the irregular samples as a missing data imputation problem. It is capable of imputing missing observations and inferring the graph at the same time. However, it only supports single time series. We use the authors' opensourced code (`https://github.com/jarrycyx/unn`).

- Rhino: Gong et al. (2022) proposed one of the most flexible SEM-based temporal structure learning framework that is capable of modelling (1) lagged parents; (2) history-dependent noise and (3) instantaneous effects. Many SEM-based structure learning approach can be regarded as a special case of Rhino. From the discussion in section 3.2, *SCOTCH* can be regarded as a continous-time version of Rhino. We use the authors' opensourced implementation (`https://github.com/microsoft/causica/tree/v0.0.0`).

- NGM: NGM (Bellot et al., 2022) proposed to use NeuralODE to learn the mean process of the SDE. Since this is the only baseline we are aware of in terms of structure learning under continuous time, this will be used as our main comparison. We use the authors' opensourced code (`https://github.com/alexisbellot/Graphical-modelling-continuous-time`).

NGM and CUTS are originally designed for single time series setup and cannot handle multiple time series. For fair comparison, we modify them by concatenating the multiple time series into a single one. That is, given $n$ time series $\{\boldsymbol{X}^{(n)}\}_{n=1}^N$ with observation times $t_1, ..., t_I$, we convert them into a single time series with observation times in $[(n - 1) * t_I + t_1, n * t_I]$ for the $n^{\text{th}}$ time series. Our assumption is that since their learning routines are batched across time points, and the concatenation points are rarely sampled, this should have small impact to the performance in comparison to the benefit of additional data. Empirically, this approach indeed improves the performance over simply selecting a single time series.

For VARLiNGaM, PCMCI, and Rhino, which cannot handle irregularly sampled data, we use zero-order hold (ZOH) to impute the missing data, which has been found to perform competitively (Cheng et al., 2023) with other imputation methods such as GP regression and GRIN (Cini et al., 2022).

### C.1 COMPARISON TO ODE-BASED STRUCTURE LEARNING

In this section, we present an extended version of the example failure case of NGM presented in section 3.2. Bellot et al. (2022) proposed a structure learning method (NGM) for learning from a single time series generated from a SDE. Their approach learns a neural ODE $d\boldsymbol{Z}(t) = \boldsymbol{f}_\theta(\boldsymbol{Z}(t))dt$ that models the mean process of the SDE and extract the graphical structure from the first layer of $\boldsymbol{f}_\theta$. Given a single observed trajectory $\boldsymbol{X} = \{\boldsymbol{X}_{t_i}\}_{i \in [I]}$, they assume that the observed data follows a multivariate Gaussian distribution $(\boldsymbol{X}_{t_1}, .. \boldsymbol{X}_{t_I}) \sim \mathcal{N}((\boldsymbol{Z}_{t_1}, .. \boldsymbol{Z}_{t_I}), \Sigma)$ with mean process $\boldsymbol{Z}_t$ given by the deterministic mean process (ODE), and a diagonal covariance matrix $\Sigma \in \mathbb{R}^{I \times I}$. As such, NGM optimizes the following squared loss:

$$\sum_{i=1}^{I} \|\boldsymbol{X}_{t_i} - \boldsymbol{Z}_{t_i}\|_2^2 \tag{35}$$

Like *SCOTCH*, NGM attempts to model the underlying continuous-time dynamics and can naturally handle irregularly sampled data. However, the Gaussianity assumption only holds when the underlying SDE is linear; that is, SDEs of the form $d\boldsymbol{X} = (\boldsymbol{a}(t)\boldsymbol{X} + \boldsymbol{b}(t))dt + \boldsymbol{c}(t)d\boldsymbol{W}_t$. For general SDEs where the drift and/or diffusion functions are nonlinear functions of the state, the joint distribution can be far from Gaussian, leading to model misspecification, resulting in the incorrect drift function even if the neural network $\boldsymbol{f}_\theta$ has the capacity to express the true drift function.

Another drawback of learning an ODE mean process using the objective in Equation 35 is that it is difficult to generalise to correctly learn from multiple time series, which can be important for recovering the underlying SDEs in practice since a single time series is just a one trajectory sample from the SDE, and thus cannot represent the trajectory multimodality due to stochasticity. In particular, simply computing a batch loss over all time series $\sum_{n=1}^{N} \sum_{i=1}^{I} \|\boldsymbol{X}_{t_i}^{(n)} - \boldsymbol{Z}_{t_i}\|_2^2$ may fail to recover the underlying dynamics when learning from multiple time series. To demonstrate the above argument, we propose a bi-modal failure case. Consider the following 1D SDE:

$$dX = Xdt + 0.01dW_t \tag{36}$$

where the trajectory will either go upwards or downwards exponentially (bi-modality)

In Figure 2a we show trajectories sampled from this SDE, where the initial state is set to $X_0 = 0$ for all trajectories. The optimal ODE mean process in terms of (batched) squared loss is given by $dZ = 0dt$, whose solution is given by the horizontal axis; in particular, while true graph by definition contains a self-loop, the inferred graph from this ODE has no edges. In Figure 2b we show the ODE mean process $\boldsymbol{f}_\theta$ learned by NGM, together with trajectory samples from the corresponding SDE $dX = \boldsymbol{f}_\theta(X)dt + 0.01dW_t$. The learned ODE mean process (in black) is close to the horizontal axis (note the scale of the vertical axis), with trajectories that do not match the data. On the other hand, in Figure 2c we see that *SCOTCH* successfully learns the underlying SDE with trajectories closely matching the observed data and demonstrating the bi-modal behavior.

## D EXPERIMENTS

### D.1 CHOICE OF SDE SOLVER

There are several choices that can affect the accuracy of the SDE solver used for *SCOTCH*. Firstly, discretization step size is an important factor of the solver; a smaller step size generally leads to a more accurate SDE solution, but at the cost of additional time and space complexity. The computational cost (with default Euler discretization) should scale inversely w.r.t. the step size. In the following, we conducted an initial verification run for the Ecoli1 dataset with half of the original step size reported below in appendix appendix D.5. Appendix D.1 compares the performance with different step sizes. We can see that $\Delta t = 0.05$ results in similar performance compared to $\Delta t = 0.025$ (while being 2x faster). Therefore, we decide to use the step size $\Delta t = 0.05$. Secondly, we chose to use a pathwise gradient estimator rather than the adjoint method (Li et al., 2020), as we found this was more efficient time-wise and we did not run into space limitations. Although theoretically, they should give the same performance, in practice, the pathwise gradient estimator may have an advange that computing its gradient does not require solving another SDE, which is subject to the accuracy of the SDE solver. It is also possible to use higher-order numerical solvers such as the Milstein method; however we did not explore this thoroughly in this work.

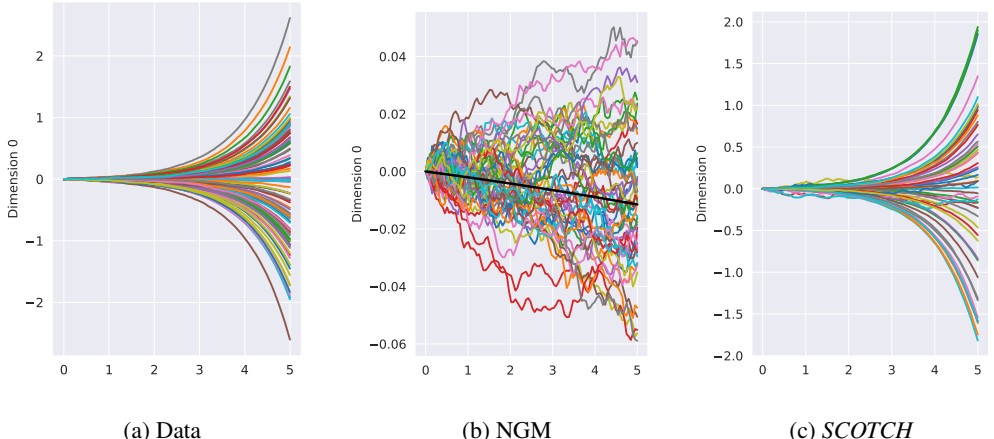

|   |   |   |
|---|---|---|
| (a) Data | (b) NGM | (c) *SCOTCH* |

Figure 2: Comparison between NGM and *SCOTCH* for simple SDE (note vertical axis scale)

|  | AUROC |
|---|---|
| $\Delta t = 0.025$ | **0.747±0.005** |
| $\Delta t = 0.05$ | **0.752±0.008** |

Table 4: Performance comparisons between different choice of discretization step size $\Delta t$ for SDE solver.

### D.2  COMPARISON TO LATENT SDEs

Though appealing at first glance, attempting to directly extract graphical structure from SDEs learned using existing methods, such as that of Li et al. (2020), is very challenging. Firstly, to extract the signature graph, one would have to evaluate the partial derivative of the drift and diffusion networks at every input point in the input domain, which is not practical. Secondly, the learned drift and diffusion functions may have different graphs, and it is unclear how we should combine these. Thirdly, there are no theoretical results to justify this approach (prior to our paper's theory). For these reasons, prior work does not admit an easy way to extract structure.

In order to construct an simple empirical baseline following this strategy, we follow the setup of Li et al. (2020), and implement each output dimension of the drift and diffusion functions as a separate neural network, i.e.

$$\boldsymbol{f} = [f_1, ..., f_D]^T, \boldsymbol{g} = [g_1, ..., g_D]^T \tag{37}$$

Using e.g. $\boldsymbol{A}_{g_j}$ to denote the weight matrix of the first layer of $g_j$, and $\boldsymbol{A}_{g_j}^k$ to denote the $k^{\text{th}}$ column of that matrix (corresponding to the $k^{\text{th}}$ input dimension, then we define:

$$\boldsymbol{H}_{k,j} = \max(|\boldsymbol{A}_{f_j}^k|_2, |\boldsymbol{A}_{g_j}^k|_2) \tag{38}$$

| Method | AUROC |
|---|---|
| PCMCI+ | $0.530 \pm 0.002$ |
| NGM | $0.611 \pm 0.002$ |
| CUTS | $0.543 \pm 0.003$ |
| Rhino | $0.685 \pm 0.003$ |
| SCOTCH | $\mathbf{0.752 \pm 0.008}$ |
| LSDE | $0.496 \pm 0.021$ |

Table 5: Performance comparison between methods on DREAM3 Ecoli1 dataset. LSDE refers to latent SDE + extracting first layer weights.

to be our (weighted) estimate of the graph structure. This has the property that whenever $\boldsymbol{H}_{k,j} = 0$, then $\frac{\partial f_j}{\partial x_k} = 0$ and $\frac{\partial g_j}{\partial x_k} = 0$. This can be extracted from a learned SDE, and we can compute an AUROC using the weights as confidence scores.

Table 5 shows results for this approach (which we call LSDE) in comparison with SCOTCH and other baselines on the DREAM3 Ecoli1 dataset. It can be seen that LSDE performs no better than random guessing at identifying the correct edges.

## D.3 Synthetic datasets: Lorenz

### D.3.1 Data generation

For the Lorenz dataset, we simulate time-series data according to the following SDE based on the $D$-dimensional Lorenz-96 system of ODEs:

$$dX_{t,d} = ((X_{t,d+1} - X_{t,d-2})X_{t,d-1} - X_{t,d})dt + F + \sigma dW_{t,i} \tag{39}$$

where $X_{t,-1} := X_{t,D-1}, X_{t,0} := X_{t,D}$, and $X_{t,D+1} := X_{t,1}$, with parameters set as $F = 10$ and $\sigma = 0.5$. We generate $N = 100$ $10-$dimensional time series, each with length $I = 100$, which are sampled with time interval 1 starting from $t = 0$ (that is, $t_1 = 0, t_2 = 1, ..., t_{100} = 99$). The initial state $X_{0,i}$ is sampled from a standard Gaussian. To simulate the SDE, we use the Euler-Maruyama scheme with step-size $dt = 0.005$.

For this synthetic dataset, we do not add observation noise to the generated time series.

To produce the irregularly sampled versions of the Lorenz dataset, for each time $t = 0, ..., 99$, we randomly drop the observed data at that time with probability $p$, independently at each time $t$ (and for all time series $n = 1, ...100$). We test using $p = 0.3, 0.6$ in our experiments.

### D.3.2 Hyperparameters

***SCOTCH*** We use Adam (Kingma & Ba, 2014) optimizer with learning rate 0.003 and 0.001 for $p = 0.3$ and $p = 0.6$, respectively. We set the $\lambda_s = 500$ and EM discretization step size $\Delta = 1$ for SDE integrator, which coincides with the step size in the data generation process. The time range is set to $[0, 100]$. We enable the residual connections for prior drift and diffusion network. We also adopt a learning rate warm-up schedule, where we linearly increase the learning rate from 0 to the target value within 100 epochs. We do not mini-batch across the time series. We train 5000 epochs for convergence.

***NGM*** We use the same hyperparameter setup as NGM (Bellot et al., 2022) where we set 0.1 for the group lasso regularizer and the learning rate as 0.005. We train NGM for 4000 epochs in total (2000 for the group lasso stage and 2000 for the adaptive group lasso stage).

***VARLiNGaM*** We set the lag to be the same as the ground truth $lag = 1$, and do not prune the inferred adjacency matrix.

***PCMCI+*** We use *partial correlation* as the underlying conditional independence test. We set the maximum lag at 2, and let the algorithm itself optimise the significance level. We use the threshold 0.07 to determine the graph from the inferred value matrix.

***CUTS*** We use the authors' suggested hyperparameters (Cheng et al., 2023) for the Lorenz dataset.

***Rhino*** We use hyperparameters with learning rate 0.01, 70 epochs of augmented lagrangian training with 6000 steps each, time lag of 2, sparsity parameter $\lambda_s = 5$, and enable instantaneous effects.

### D.3.3 Additional results

Figure 3 shows the curve of other metrics.

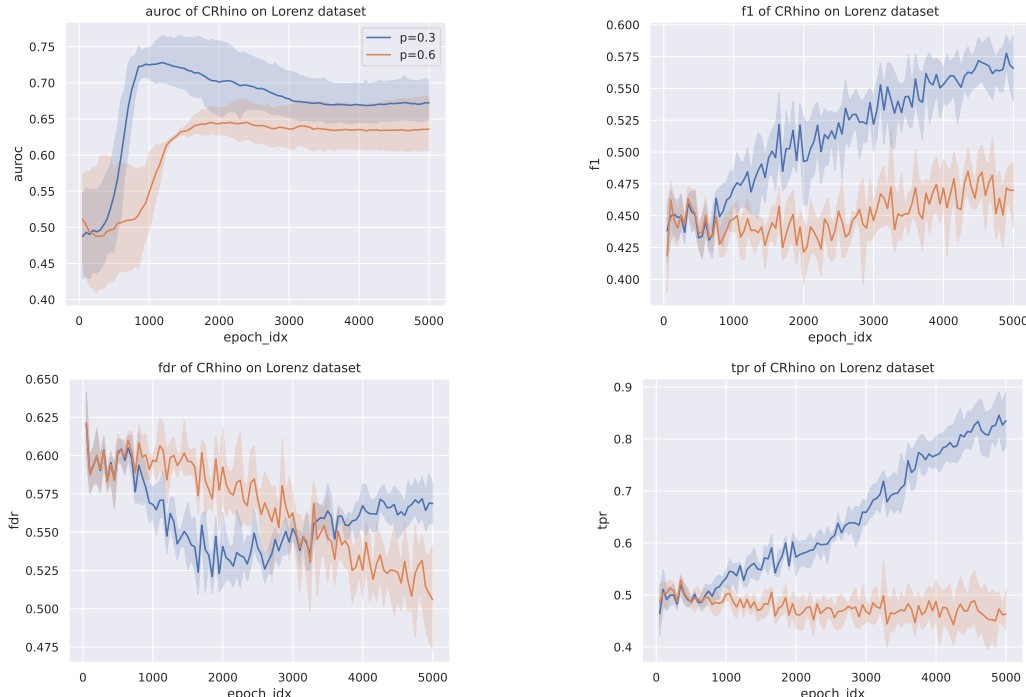

Figure 3: The AUROC (top left), F1 score (top right), false discovery rate (bottom left) and true positive rate (bottom right) curves of *SCOTCH* for Lorenz dataset. The shaded area indicates the $95\%$ confidence intervals. Blue color indicates the dataset with missing probability 0.3 and orange color indicates missing probability 0.6.

## D.4 SYNTHETIC DATASETS: GLYCOLYSIS

### D.4.1 DATA GENERATION

In this synthetic experiment, we generate data according to the system presented by Daniels & Nemenman (2015), which models a glycolyic oscillator. This is a $D = 7$ dimensional system with the following equations:

$$dX_{t,1} = \left(2.5 - \frac{100X_{t,1}X_{t,6}}{1 + (X_{t,6}/0.52)^4}\right) dt + 0.01dW_{t,1}$$

$$dX_{t,2} = \left(\frac{200X_{t,1}X_{t,6}}{1 + (X_{t,6}/0.52)^4} - 6X_{t,2}(1 - X_{t,5}) - 12X_{t,2}X_{t,5}\right) dt + 0.01dW_{t,2}$$

$$dX_{t,3} = (6X_{t,2}(1 - X_{t,5}) - 16X_{t,3}(4 - X_{t,6})) dt + 0.01dW_{t,3}$$

$$dX_{t,4} = (16X_{t,3}(4 - X_{t,6}) - 100X_{t,4}X_{t,5} - 13(X_{t,4} - X_{t,7})) dt + 0.01dW_{t,4}$$

$$dX_{t,5} = (6X_{t,2}(1 - X_{t,5}) - 100X_{t,4}X_{t,5} - 12X_{t,2}X_{t,5}) dt + 0.01dW_{t,5}$$

$$dX_{t,6} = \left(-\frac{200X_{t,1}X_{t,6}}{1 + (X_{t,6}/0.52)^4} + 32X_{t,3}(4 - X_{t,6}) - 1.28X_{t,6}\right) dt + 0.01dW_{t,6}$$

$$dX_{t,7} = (1.3(X_{t,4} - X_{t,7}) - 1.8X_{t,7}) dt + 0.01dW_{t,7}$$

As with the Lorenz dataset, we simulate $N = 100$ time series of length $I = 100$, starting at $t = 0$ and with time interval 1. The initial state is sampled uniformly from the ranges $X_{0,1} \in [0.15, 1.60], X_{0,2} \in [0.19, 2.16], X_{0,3} \in [0.04, 0.20], X_{0,4} \in [0.10, 0.35], X_{0,5} \in [0.08, 0.30], X_{0,6} \in [0.14, 2.67], X_{0,7} \in [0.05, 0.10]$, as indicated in Daniels & Nemenman (2015). To simulate the SDE, we use the Euler-Maruyama scheme with step-size $dt = 0.005$.

For this synthetic dataset, we do not add observation noise to the generated time series.

### D.4.2 HYPERPARAMETERS

*SCOTCH*   We use the same hyperparameter as Lorenz experiments. The only differences are that we use learning rate $0.001$ and set $\lambda_s = 200$. We train *SCOTCH* for 30000 epochs for convergence.

**NGM**   Since Bellot et al. (2022) did not release the hyperparameters for their glycolysis experiment, we use the default setup in their code. They are the same as the hyperparameters in Lorenz experiments.

**VARLiNGaM**   Same as Lorenz experiment setup.

**PCMCI+**   Same as Lorenz experiment setup.

**CUTS**   Same as Lorenz experiment setup.

**Rhino**   Same as Lorenz experiment setup.

### D.4.3 ADDITIONAL RESULTS

Table 6 shows the performance comparison of *SCOTCH* to NGM with the original glycolysis data, where the data have different variable scales. We can observe that this difference in scale does not affect the AUROC of *SCOTCH* but greatly affects NGM. Since AUROC is threshold free, we can see that *SCOTCH* is more robust in terms of scaling compared to NGM. A possible reason is that the stochastic evolution of the variables in SDE can help stabilise the training when encountering difference in scales, but ODE can easily overshoot due to its deterministic nature.

Figure 4 shows the curves of different metrics. Interestingly, we can see that data normalisation does not improve the AUROC performance (compared to NGM), but does increase the f1 score. This may be because f1 is threshold sensitive and the default threshold of $0.5$ might not be optimal. We can see this through the TPR plot, where "Original" has very low value.

Table 6: Performance comparison with original Glycolysis data

|         | AUROC            | TPR ↑            | FDR ↓            |
|---------|------------------|------------------|------------------|
| *SCOTCH* | **0.7352±0.019** | **0.3623±0.007** | **0.1575±0.05**  |
| NGM     | 0.5248±0.057     | 0.3478±0.035     | 0.4559±0.094     |

## D.5 DREAM3 DATASET

In this appendix, we will include experiment setups, hyperparameters and additional plots for Dream3 experiment.

### D.5.1 HYPERPARAMETERS

*SCOTCH*   We follow similar setup as Lorenz experiment. The differences are that the learning rate is $0.001$. The time range is set to $[0, 1.05]$ with EM discretization step size $0.05$, which results in exactly 21 observations for each time series. We choose sparisty coefficient $\lambda_s = 200$. For all sub-datasets, we normalize the data to have $0$ mean and unit variance for each dimension. We use the above hyperparameters for Ecoli1, Ecoli2 and Yeast1 sub-datasets. For Yeast2, we only change the learning rate to be $0.0005$. For Yeast3, we change the $\lambda_s = 50$. We train *SCOTCH* for 30000 epochs until convergence.

**NGM**   For NGM, we follow the same hyperparameter setup as (Cheng et al., 2023), where we set the group lasso regulariser as $0.05$, learning rate $0.005$. We train NGM with $4000$ epochs (2000 each for group lasso and adaptive group lasso stages). For fair comparison, we use the same observation time (i.e. equally spaced time points within $[1, 1.05]$ and step size $0.05$).

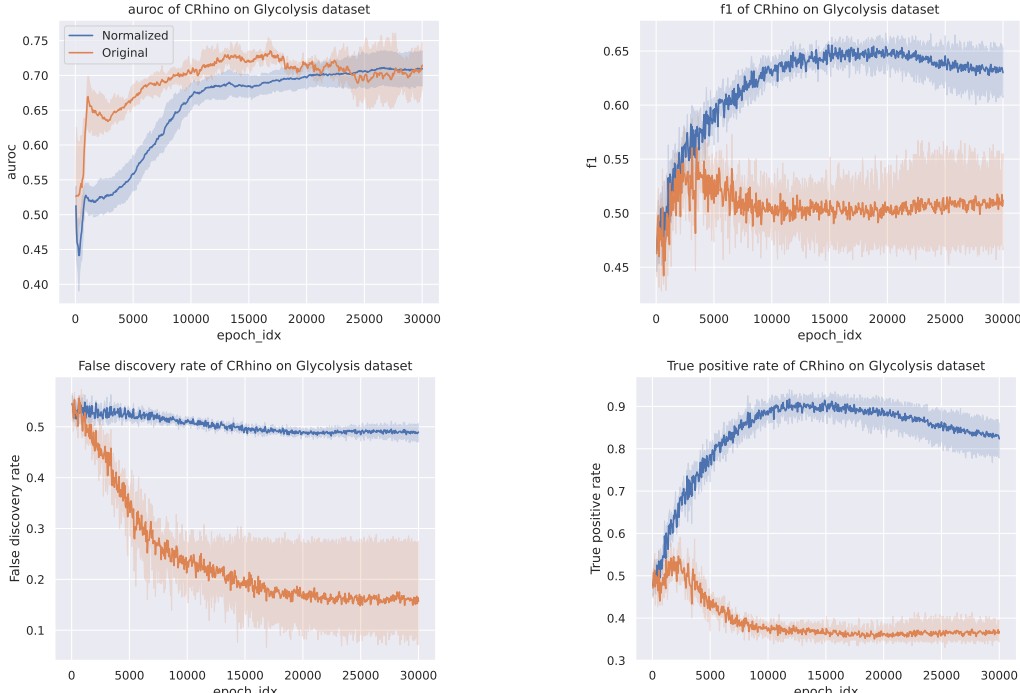

Figure 4: The AUROC (top left), F1 score (top right), false discovery rate (bottom left) and true positive rate (bottom right) curves of *SCOTCH* for Glycolysis dataset. The shaded area indicates the 95% confidence intervals. Blue color indicates the normalized dataset and orange color indicates the original dataset.

**PCMCI+ and Rhino** As the experiment setup is the same, we directly cite the number from Gong et al. (2022).

**CUTS** We use the authors' suggested hyperparameters (Cheng et al., 2023) for the DREAM3 datasets.

### D.5.2 ADDITIONAL PLOTS

In this section, we include additional metric curves of *SCOTCH* in fig. 5. Each curve is obtained by averaging over 5 runs and the shaded area indicates the 95% confidence interval. From the value of f1 score, FDR and TPR, we can see DREAM3 is indeed a challenging dataset, where all f1 scores are below 0.5 and FDR only drops to 0.7. From the TPR plot, it is expected to drop at the beginning and then increase during training, which is the case for Ecoli1, Ecoli2 and Yeast1. TPR corresponds well to AUROC and F1 score, since Ecoli1, Ecoli2 and Yeast1 have much better values compared to Yeast2 and Yeast3.

### D.6 NETSIM

### D.6.1 EXPERIMENT SETUP

For the Netsim dataset, we generate the missing data versions in the same way as the Lorenz dataset (see appendix D.3).

### D.6.2 HYPERPARAMETERS

**SCOTCH** We use similar hyperparameter setup as Dream3 (appendix D.5.1), but we change $\lambda_s = 1000$ and use the raw data without normalisation. We train *SCOTCH* for 10000 epochs.

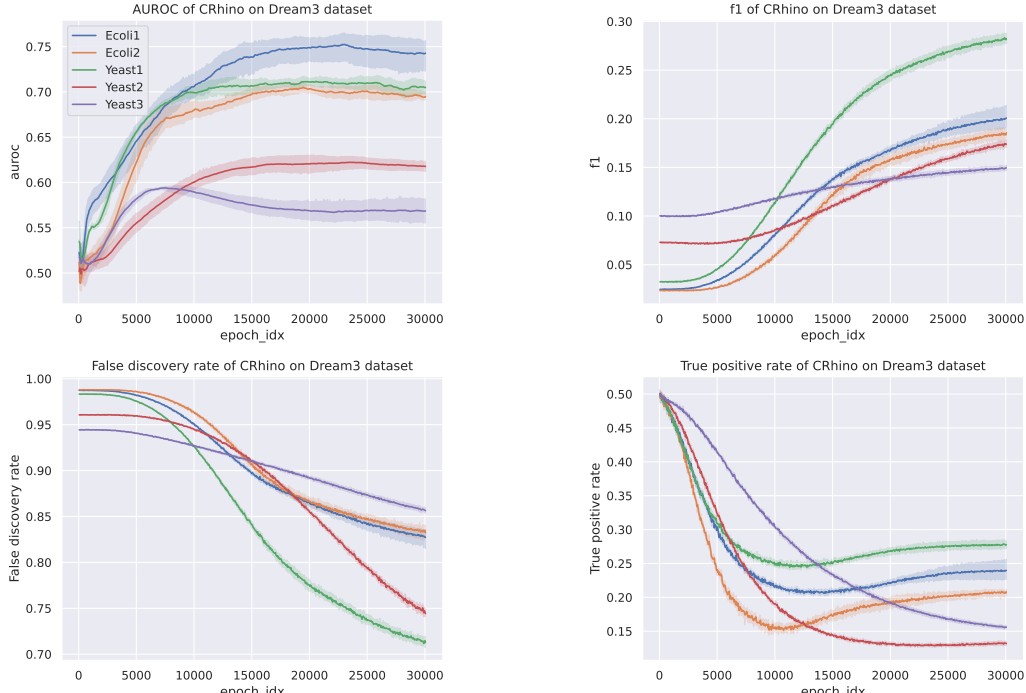

Figure 5: The AUROC (top left), F1 score (top right), false discovery rate (bottom left) and true positive rate (bottom right) curves of *SCOTCH* for each DREAM3 sub-datasets. The shaded area indicates the 95% confidence intervals.

**NGM**   We follow the same setup as DREAM3 experiment, which also coincides with the setup used in Cheng et al. (2023).

**PCMCI**   We follow the same setup as Lorenz and use threshold 0.07 to infer the graph.

**CUTS**   We use the authors' suggested hyperparameters (Cheng et al., 2023) for the Netsim dataset.

**Rhino and Rhino+NoInst**   We directly cite the number from Gong et al. (2022) for the full dataset, and use the same hyperparameters as Gong et al. (2022) for both $p = 0.1$ and $p = 0.2$ Netsim datasets.

### D.6.3   ADDITIONAL PLOTS

We include additional metric curves of *SCOTCH* on Netsim dataset in fig. 6. From the plot, we can see Netsim is a easier dataset compared to DREAM3 since the dimensionality is much smaller. An interesting observation is f1 score does not necessarily correspond well to auroc since f1 score is threshold dependent (by default we use 0.5) but not auroc. To evaluate the robustness of the model, we decide to report AUROC instead of f1 score.

## E   INTERVENTIONS

Aside from learning the graphical structure between variables, one might also be interested in analysing the effect of applying external changes, or interventions, to the system. Broadly speaking, there are two types of interventions that we can consider in a continuous-time model. The first is to intervene on the dynamics (that is, the drift or diffusion functions), possibly for a set period of time. The second is to directly intervene on the value of (some subset of) variables. The goal is to employ our learned *SCOTCH* model in order to predict the effect of these interventions on the underlying system.

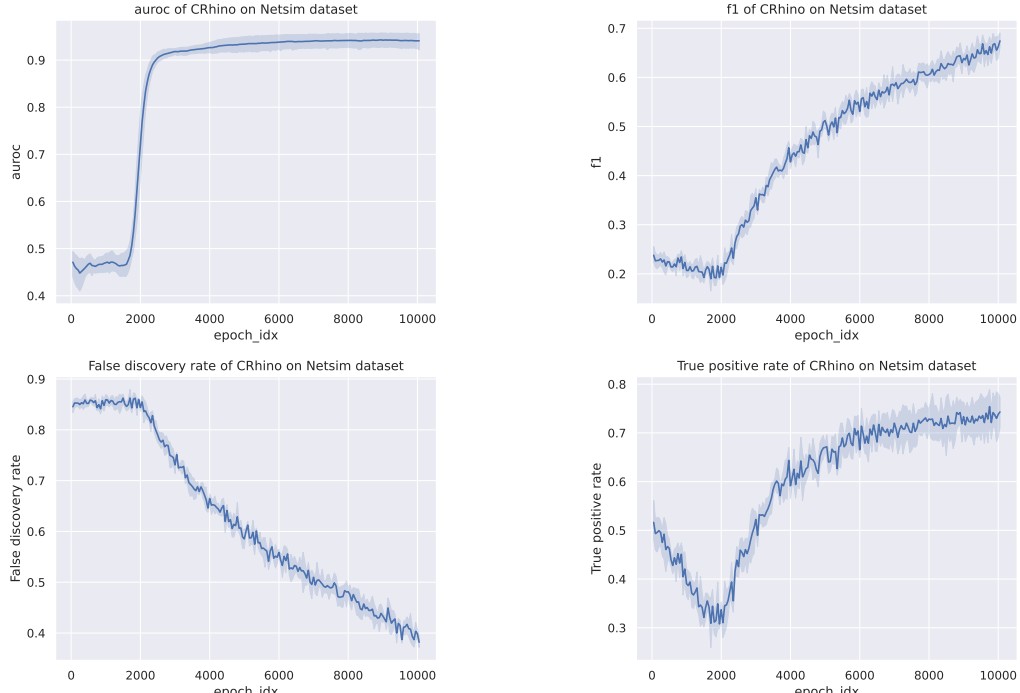

Figure 6: The AUROC (top left), F1 score (top right), false discovery rate (bottom left) and true positive rate (bottom right) curves of *SCOTCH* for Netsim dataset. The shaded area indicates the 95% confidence intervals.

The former is easy to implement as we need only replace (parts of) the learned drift/diffusion function with the intervention. However, the latter is slightly more subtle than it might first appear. (Hansen & Sokol, 2014) proposed to define such an intervention as a function that fixes the value of a particular variable as a function of the other variables. However, it is unclear how we can generalize this to interventions affecting more than one variable. For example, a intervention policy $Z_1 \leftarrow Z_2, Z_2 \leftarrow Z_1 + 1$ creats a feedback loop whose semantics are not easy to resolve. Thus, we propose the following definition:

**Definition 3** (State-space Intervention). *Given a $D$-dimensional SDE, a state-space intervention is an idempotent function $\iota(t, \mathbf{Z}) : \mathbb{R}^{D+1} \to \mathbb{R}^D$; that is, $\iota(t, \iota(t, \mathbf{Z})) \equiv \iota(t, \mathbf{Z})$. The corresponding intervened stochastic process is defined by:*

$$\tilde{\mathbf{Z}}_t = \iota \left( t, \tilde{\mathbf{Z}}_0 + \sum_{d \in [D]} \int_0^t \mathbf{f}(\tilde{\mathbf{Z}}_s) ds + \sum_{d \in [D]} \int_0^t \mathbf{g}(\tilde{\mathbf{Z}}_s) d\mathbf{W}_s \right) \tag{40}$$

The requirement of idempotence captures the intuition that applying the same intervention twice should result in the same result. Some examples of interventions are given as follows:

- **Identity**: If $\iota(t, \mathbf{Z}) = \mathbf{Z} \ \forall t \in [T_1, T_2], \mathbf{Z} \in \mathbb{R}^D$, then the system evolves accoridng to the original SDE in this time period, with initial state $\tilde{\mathbf{Z}}_{T_1}$.

- **Ordered Intervention**: Given some ordered subset of the variables, we can consider intervening on each variable in order, as a function of the previous variables in the order. That is, we restrict each dimension $\iota_i$ of the intervention output to be of the form

$$\iota_i(t, \mathbf{Z}_t) = \iota_i(t, \mathbf{Z}_{t, <i}) \tag{41}$$

where $\mathbf{Z}_{t, <i} = \{\mathbf{Z}_{t,j} : j < i\}$. It can easily be seen that $\iota$ is always idempotent in this case.

- **Projection**: Another example of an idempotent function is a projection. This could simulate a setting where external force is applied to ensure the SDE trajectories satisfy spatial

constraints. Note that a projection cannot necessarily be expressed as an ordered intervention (e.g. consider projection onto a sphere).

In practice, we implement state-space interventions in SDEs learned from *SCOTCH* by modifying the SDE solver (e.g. Euler-Maruyama) such that each step is followed with an intervention assignment $\boldsymbol{Z}_t \leftarrow \iota(t, \boldsymbol{Z}_t)$.