# OpenReview forum: "Neural structure learning with stochastic differential equations"
_ICLR.cc/2024/Conference — ICLR 2024 poster_

### Official Review · Reviewer_gu2c · 2023-10-28

**Soundness:** 3 good
**Presentation:** 3 good
**Contribution:** 2 fair
**Rating:** 6
**Confidence:** 3

**Summary:**

The authors describe a type of latent variable model that can be identified from the high dimensional observations. Specifically, the authors describe a type of stochastic differential equation based model where the parameters of the drift and diffusion are given by some graph. The graph induces a particular type of interaction between the variables and one can then estimate some causal structures from this. The authors then provide a series of experiments demonstrating the applicability of their proposed method. They additionally theoretically analyze the structural identifiability in terms of the graph structure.

**Strengths:**

The topic is important and the authors consider an important class of stochastic processes to work on. The method empirically performs well in comparison to existing models.  The method is additionally well motivated with a few theoretical results on identifying the latent structure as well as the parameters of the diffusion. The interpretation of the drift and diffusion in terms of graphs provides an easier interpretation of the dependencies of the variables.

**Weaknesses:**

Aside from the graph interpretation if the drift/diffusion, the work is similar to a lot of existing works, so I wonder a bit how it is practically different from some of the others. For example, some of the theoretical results on latent identifiability are from the work Hasan et al but the authors do not consider that work as a baseline or mention it within the related work, though they estimate similar quantities.  Additionally the authors mention that this is a novel latent SDE formulation but most of the methodologies follow existing methods using a change of measure (e.g. Li et al among others).

The numerical results were applied to fairly low dimensional datasets though it seems like the work could be applied to higher dimensional settings (as some of the related work applied to higher dimensional settings).

To summarize, I think I'm mainly confused as to what is different in this work compared to existing works since it's methodologically similar and the theoretical results are also similar. From what I've understood, it's mainly that the graph structure dictating the interaction is explicit.

**Questions:**

What is the main difference with other latent SDE models that have been proposed? Is it that the drift/diffusion are factorized in terms of the graph implying a particular architecture for those functions?

If one uses existing methods for uncovering SDEs (that use, for example, neural networks to represent functions) is there a way to estimate the graph structure from the learned function (e.g. by computing the partial derivatives between the function and the input)? How would the methods compare if the authors applied this technique to existing methods that estimate latent SDEs?

Do the authors have an idea of how the performance scales to higher dimensional observations?

---

> ### Author Response · Authors · 2023-11-17
> **Author's Response 1**
>
> We thank the reviewer for their hard work and helpful feedback.
>
> > "What is the main difference with other latent SDE models that have been proposed? ..."
>
> The main conceptual difference in comparison to prior work on latent SDEs is that we tackle the problem of structure learning, or in other words, the qualitative dependencies between variables in the SDE. In this respect, our theoretical results are crucial for showing when this can be achieved (regardless of the specifics of the SDE learning method). This complements existing results on structural identifiability in the discrete-time structure learning literature, as well as the ODE structural identifiability of Bellot et al. (2022).
>
> In more detail, Theorem 4.1 is a new result that shows (under assumptions 1, 2) the identifiability of the graphical structure when observing data generated directly from the corresponding SDEs. Then, Theorem 4.2 leverages the proof of Hasan et al. (as stated in our proof), and shows the identifiability of the graph (and SDE) even when the SDE is over latent variables, if the time range is bounded with infinite data or the observation interval is fixed. On the other hand, Hasan et al. focus on the identifiability of the model with fixed time intervals and cannot show the identifiability of the graph since they lack Theorem 4.1.
>
> Finally, Theorem 4.3 shows the consistency of our particular method, SCOTCH, in recovering the correct graph in the limit of infinite data, justifying its use for structure learning. This consistency in continuous time has not been shown in Hasan et al.
>
> We should also emphasize that although Hasan et al. presented its framework under SDEs, its nature is a discrete model that requires observation at fixed time intervals. This is because they assume an Euler discretization of the SDE so that the corresponding model is a discrete state-space model. On the other hand, our model is much more general and inherently continuous in time, not restricted to fixed-interval Euler discretization.
>
> > "Aside from the graph interpretation in the drift/diffusion, the work is similar to a lot of existing works, so I wonder a bit how it is practically different from some of the others"
>
> In addition to the conceptual and theoretical results above, we now aim to clarify the practical/methodological distinctions. In short, SCOTCH share some similarities to e.g. Li et al. in terms of the SDE formulation, but with some important differences:
>
> 1. **Bayesian Structure Learning**: In SCOTCH, we employ Bayesian structure learning and explicitly model a variational posterior distribution over graphs. Practically, this means that we sample from this distribution during training, effectively "masking" the learned neural network drift and diffusion functions in different ways; this provides a signal to learn the structure. On the other hand, Li et al. treated the model parameters as a point estimate and do not have the concept of the graph. Even if one tries to extract the "graph" from the weights, it is a point-estimate graph rather than a distribution over graphs.
> 2. **Objective Function and Network Design**: The incorporation of a learnable graph distribution leads to a distinct (though related) ELBO (Eq. 15) compared to Li et al. Correspondingly the network designs are also different; for example, the prior drift and diffusion functions follow the design of Eq. 8, while the posterior drift function $h_{\psi}$ takes as input the graph as explained in Section 3.1.
> 3. **Latent Dimensionality**: A more straightforward difference is that SCOTCH always has the same latent and observed dimensionality, unlike other works. This ensures that the graphical structure of the latent system can be identified from observed data (under assumptions).

---

> ### Author Response · Authors · 2023-11-17
> **Author's Response 2**
>
> > "If one uses existing methods for uncovering SDEs ... is there a way to estimate the graph structure from the learned function (e.g. by computing the partial derivatives between the function and the input)?"
>
> Attempting to extract a graph from the learned SDE in this way, without the explicit concept of a SDE graph, is challenging for a few reasons.
>
> Firstly, to use partial derivatives, one would have to evaluate the partial derivative of the drift and diffusion networks at every input point in the input domain, which is not practical. Secondly, the learned drift and diffusion functions may have different graphs, and it is unclear how we should combine these. Thirdly, there are no theoretical results to justify this approach (prior to our paper's theory). For these reasons, prior latent SDE work (e.g. Li et al.) does not admit an easy way to extract structure.
>
> | Method | AUROC |
> | --- | --- |
> | PCMCI+ | 0.530 $\pm$ 0.002 |
> | NGM | 0.611  $\pm$ 0.002 |
> | CUTS | 0.543  $\pm$ 0.003 |
> | Rhino | 0.685   $\pm$ 0.003|
> | SCOTCH | **0.752 $\pm$ 0.008** |
> | LSDE | 0.570 |
>
> In order to construct an empirical baseline for comparison, one way in which the graph can be extracted from the latent SDE is if the each output dimension of the drift and diffusion functions is implemented by a different neural network. Then we can simply extract the weights of the first layer of the networks, following the approach of (Bellot et al. 2022) for ODEs. As shown in the table, this approach (which we call LSDE) performed reasonably compared to some of the other baselines, but far worse than SCOTCH. We have added this to the appendix and provide more details there.
>
> > "Do the authors have an idea of how the performance scales to higher dimensional observations?"
>
> In our experiments, we do test on the 100-dimensional DREAM3 dataset. This is a very challenging dataset due to its high dimensionality with small number of observations (below 1000 observations). However, our model significantly outperforms the baselines, and to the best of our knowledge, SCOTCH achieves the state-of-the-art performance. In addition, as far as we are aware, this is the highest dimensionality dataset tackled among the related works, both regarding structure learning and SDE learning. Hasan et al. does consider modelling MNIST observed data, but the latent dimensionality (upon which the SDE operates) is very small, consisting of only digit positions.
>
>
> If we have addressed your concerns to your satisfaction, we would be grateful if you would consider increasing your score. We are of course happy to clarify further or answer any additional questions you may have.

---

> ### Comment · Area_Chair_qPRR · 2023-11-22
> **Please acknowledge author response**
>
> Please acknowledge author replies, and in particular, indicate whether your concerns have been addressed or require further discussion.

---

> > ### Comment · Reviewer_gu2c · 2023-11-23
> >
> > Thanks to the authors for their response and the additional context. The comment on using partial derivatives is helpful in seeing why the explicit addition of the graph is necessary as well as the experiment. The authors also largely confirmed what I initially found to be the difference between the relevant works. I now have a better understanding of the contribution, and I'll increase the score, though I do not quite see the significance the identifiability of the graph structure when in context of existing identifiability results on parameters of SDEs. This seems like specific application where the drift has an explicit graph structure. One other thing I did not mention is it may be a good idea to take another edit on the appendix and proofs since they are a bit hard to follow/contain misspellings (e.g. "supermum" -> supremum, "manupilating" -> manipulating).

---

> > > ### Author Response · Authors · 2023-11-23
> > >
> > > Thanks for the valuable feedbacks from the reviewer. We will take another edit on the appendix and proofs to fix the typos and grammar errors.
> > >
> > > The identifiability results are important and necessary for structure learning and the methods from Hasan et al cannot be applicable for our settings due to the following reason:
> > >
> > > - All the theory of Hasan et al requires a special discretisation, EM discretisation, and needs two strong assumptions: (i) for every discretisation step, it need the corresponding observation; (ii) they assume an isotropic diffusion function. However, our identifiability does not have those constraints. Theorem 4.1 shows the identifiability of graph under continuous time regardless of the discretisation, and allows diagonal diffusion function.
> > > - The variational posterior formulation of ours is different from Hasan et al. We use a proper SDE as the approximate posterior whereas Hasan et al uses Gaussian encoder, which is similar to an VAE. Thus, this is not a trivial change since the consistency results (Theorem 6 in Hasan) cannot be applied. Therefore, we derive a new theorem 4.3 to show the consistency under SDE approximate posterior.
> > >
> > > Thus, to summarise, if we use EM discretisation, constant diffusion, Gaussian encoder, and do not have any missing data in the system, then we have a similar model as Hasan et al. But our framework is much more general in a sense that (i) you can choose SDE solver beyond EM; (ii) allow diagonal diffusion matrix; (iii) have much more flexible approximate posterior through SDE.
> > >
> > > Hope this clarifies your concerns.

---

> > > > ### Comment · Reviewer_gu2c · 2023-11-23
> > > >
> > > > Many thanks for the continued discussion and responses, I appreciate the authors' time in helping me understand the paper.
> > > >
> > > > In the proof of the continuous case, as far as I understood, the proof of Lemma A.3 uses an Euler discretization to show uniqueness of the semigroup. In that sense, the continuous case proof follows directly from an Euler discretization which makes the identifiability result in continuous time seem not as significant. I agree that it is different in that one does not require an explicit discretization, but the the proof is largely similar.
> > > >
> > > > The comment on the variational posterior seems to follow the work of Tzen and Raginsky which all follow Girsanov's theorem and having an expressive enough posterior (discussed in [1] as well). This is why I am confused about the significance of the consistency result.
> > > >
> > > > Note that my opinion on what is significant is not a reason to reject a paper, but I am mainly saying that there should be more context with respect to existing work that tackles similar problems.
> > > >
> > > > [1] Huang et al, A Variational Perspective on Diffusion-Based Generative Models and Score Matching, NeurIPS 2021

---

### Official Review · Reviewer_XYcj · 2023-11-01

**Soundness:** 3 good
**Presentation:** 2 fair
**Contribution:** 2 fair
**Rating:** 6
**Confidence:** 4

**Summary:**

The paper presents a structure learning method which aims to learn a graph fused into drift and diffusion function in stochastic differential equations (SDEs). The variational inference is followed Li et al, 2020 with additional conditions over graphs. The paper then studies structure identifiability of the model using tools from stochastic calculus. Experiments are conducted in both synthetic data sets and real-world data sets, comparing the proposed methods with alternative approaches.

**Strengths:**

- The paper gives an interesting connection between structural learning and SDE. The theory and practice from SDE literature builds a good foundation for this direction.
- There is a strong empirical evidence that SCOTCH performs well across multiple tasks.

**Weaknesses:**

Although the paper provides empirical results compared to existing models, it does not provide any analysis about obtained graphs. For me, I am more curious about the quality of produced graphs from SCOTCH compared to other models, and how we understand their structures. I wonder if there is any way to visualize the graphs.

**Questions:**

Please see weakness part.

---

> ### Author Response · Authors · 2023-11-17
> **Author's Response**
>
> We thank the reviewer for their positive assessment of our paper.
>
> > "I am more curious about the quality of produced graphs from SCOTCH compared to other models, and how we understand their structures. I wonder if there is any way to visualize the graphs."
>
> In our experiments, we do quantitatively compare the quality of the graphs to the ground truth from SCOTCH vs other models using the AUROC metric, where a higher value corresponds to a greater degree of conformity to the true graph. For visualizing the graph, one can simply sample the adjacency matrix from SCOTCH and use python packages like *networkx* to automatically plot the corresponding graphs.
>
> We hope that we have understood your concerns regarding this. Please let us know if not, or if you have any other feedback that could improve the paper - we would be very grateful.

---

### Official Review · Reviewer_pC52 · 2023-11-01

**Soundness:** 3 good
**Presentation:** 3 good
**Contribution:** 3 good
**Rating:** 8
**Confidence:** 3

**Summary:**

The paper introduces a novel structure learning method, SCOTCH, which leverages neural stochastic differential equations (SDE) and variational inference to infer posterior distributions over possible structures in continuous-time data. Traditional structure learning methods assume discrete-time processes with regularly spaced observations, which can lead to incorrect models. SCOTCH, however, is capable of handling both learning from and predicting observations at arbitrary time points. The authors establish the structural identifiability and consistency of SCOTCH under certain conditions. Empirical evaluations on synthetic and real-world datasets demonstrate its superior performance compared to relevant baselines, even with irregularly sampled data.

**Strengths:**

1. One of the primary strengths of this paper is its originality in tackling the problem of structure learning in continuous-time data. While many existing methods focus on discrete-time processes with regular observations, SCOTCH introduces a novel approach that can handle irregularly sampled data in continuous time.

2. The paper provides a strong theoretical foundation for the proposed method. The establishment of structural identifiability conditions and the proof of consistency under infinite data limits.

3. The paper maintains a high level of clarity in explaining the methodology, making it accessible to a wide audience. Additionally, the empirical evaluations conducted on both synthetic and real-world datasets demonstrate the effectiveness of SCOTCH. The comparison with relevant baselines under different sampling conditions reinforces the credibility of the proposed approach.

**Weaknesses:**

1. SCOTCH relies on neural stochastic differential equation (SDE) methods, which means that its performance and computational cost are inherently linked to the accuracy of numerical SDE solvers. The paper would benefit from a more in-depth analysis and discussion regarding the sensitivity of the proposed method to the SDE solvers.

2. The paper aims to sample the structure of G and obtain a sparse G. However, it falls short in providing a thorough discussion of the reasons behind the choice of priors for structure sampling. Additionally, there is no exploration of whether achieving sparsity in G leads to improved estimation performance, or if it primarily results in more interpretable results.

**Questions:**

See Weaknesses

---

> ### Author Response · Authors · 2023-11-17
> **Author's response**
>
> We thank the reviewer for their valuable feedback, and recognition of our contributions.
>
> > "The paper would benefit from a more in-depth analysis and discussion regarding the sensitivity of the proposed method to the SDE solvers."
>
> The reviewer is correct that there are a number of factors related to the accuracy of the SDE solution from the SDE solver. Firstly, there is the matter of discretization step size of the solver; a smaller step size generally leads to more accurate SDE solution, but at the cost of time and space complexity. The computational cost (with default Euler discretization) will scale inversely with the step size. We conducted an additional experiment for the Ecoli1 dataset (in DREAM3) with a smaller step size and added it to the appendix of the revised paper and below.
>
> | | AUROC |
> | --- | --- |
> |$\Delta t= 0.025$ | 0.747$\pm$0.005 |
> |$\Delta t= 0.05$ | 0.752$\pm$0.008 |
>
> We can see that, at least for the Ecoli1 dataset, a finer discretization does not necesarily improves the performance, while increasing computational cost. Therefore, for Dream3 dataset, we use $\Delta t=0.05$ as default.
>
> Secondly, we chose to use a pathwise gradient estimator rather than the adjoint method, as we found this was more efficient time-wise and we did not run into memory limitations. Although theoretically, they should give the same performance, in practice, the pathwise gradient estimator may have an advantage that computing its gradient does not require solving another SDE, which is subject to the accuracy of the SDE solver. Finally, it is also possible to use higher-order numerical solvers such as the Milstein method, in place of Euler-Maruyama; however we have not thoroughly explored this in the current work. We have added discussion on SDE solvers to the experimental section and appendix.
>
> > "...a thorough discussion of the reasons behind the choice of priors for structure sampling... whether achieving sparsity in G leads to improved estimation performance, or if it primarily results in more interpretable results."
>
> The reason we need the sparsity prior is because in theory, a fully connected graph will always have the greatest capacity to fit data. The purpose of incorporating a sparsity term is to avoid false positive edges, i.e. spurious edges that may slightly improve the ELBO but are not present in the underlying system. In real-world systems (e.g. DREAM3), we found this helped to more accurately predict the ground-truth graph.
>
> L1 sparsity priors (Lasso) are the most common type of prior used in structure learning, so we adopt this for our method. That said, there are other possibilities. For example, another prior sometimes used in DAG structure learning (Eggeling et al. 2019) assigns probability $p(G) \propto \prod_{d=1}^{D} \frac{1}{{D-1 \choose |pa(d)|}}$, where $pa(d)$ is the set of parents of the $d^{\textnormal{th}}$ variable. In theory we can use any graph prior that assigns positive probability to every graph. The L1 prior turns out to be particularly convenient (in combination with our variational graph posterior) as the KL-divergence in Eq. (14) is tractable.
>
> (Eggeling et al. 2019) Ralf Eggeling, Jussi Viinikka, Aleksis Vuoksenmaa, Mikko Koivisto. On Structure Priors for Learning Bayesian Networks. AISTATS 2019.

---

> > ### Comment · Reviewer_pC52 · 2023-11-23
> >
> > Thank authors for the detailed reply.

---

### Official Review · Reviewer_rM8f · 2023-11-02

**Soundness:** 3 good
**Presentation:** 3 good
**Contribution:** 4 excellent
**Rating:** 8
**Confidence:** 3

**Summary:**

This paper introduces SCOTCH  a continuous-time stochastic model that combines stochastic differential equations (SDEs) and variational inference to model temporal processes and learn the underlying graph structure of the dynamics. The results include theory on sufficient conditions for SCOTCH to be structurally identifiable and empirical results on a variety of biomedical systems

**Strengths:**

-Continuous dynamics modeling is a good extension (over RHINO etc) because of the ability to incorporate irregularly sampled time series
-The Ito diffusion model is very flexible for modeling a variety of types of dynamic
-I think that the independent diagonal noise assumption is perfectly reasonable for the generation of the dynamics
-It is impressive that this network both learns the dynamics as well as matches the flows
-The identifiability results are important despite their assumptions

**Weaknesses:**

-I think the assumptions (1, 2) should be more clearly stated (I had to comb through the text to find them)
-In specific homogenous drift and diffusion processes assumption may not be able to model some types of dynamics, some discussion on that would be interesting
-Not sure if I believe the explanation of the improved performance of RHINO on the Netsim data

**Questions:**

What is the effect of the sparsity prior on the structure? Can a different type of sparsity be used?

---

> ### Author Response · Authors · 2023-11-17
> **Author's response**
>
> We thank the reviewer for their encouraging assessment of the strengths of our paper and for their comments, which we address below.
>
> > "I think the assumptions (1, 2) should be more clearly stated"
>
> Agreed; we have updated the manuscript to place the assumptions in the main text.
>
> > "homogenous drift and diffusion processes assumption may not be able to model some types of dynamics"
>
> This is definitely an interesting point, which we now discuss in the conclusion of the revised manuscript.  In a nutshell, with homogeneous drift and diffusion functions, SCOTCH cannot model non-stationary dynamics. For example, systems with obvious changepoints (e.g. climate science models) will not have time-homogenous drift/diffusion functions. Unfortunately, our theoretical results no longer hold if time-dependent graphs are allowed.
>
> > "...the explanation of the improved performance of RHINO on the Netsim data"
>
> | Method | AUROC |
> | --- | --- |
> | cMLP | 0.93 |
> | Rhino+NoInst | 0.93 |
> | Rhino+g | 0.974 |
> | Rhino | 0.99 |
>
> With regards to the performance of SCOTCH vs Rhino, we copy (part of) Table 3 of Gong et al. (2022) above, showing that Rhino can achieve almost perfect AUROC on the Netsim dataset when allowed to use instantaneous effects, even when restricted to Gaussian noise (Rhino+g), it can achieve 0.974. Both Rhino+NoInst (no instantaneous effects), and the Granger causality method cMLP (which cannot model instantaneous effects) can only achieve 0.93 AUROC, indicating the importance of instataneous effects on this dataset. As explained in Section 3.2, SCOTCH also does not incorporate instantaneous effects, so the performance is worse compared with Rhino. We also highlight the corresponding sections in the revised paper.
>
> As for SCOTCH vs Rhino+NoInst with both complete and missing data, we can see that the performance is broadly similar and superior to other baselines. Although the performance of Rhino is slightly better than SCOTCH with missing data, we are not entirely sure why SCOTCH does not outperform Rhino+NoInst. We suspect it may be due to pecularities of the NetSim dataset, or the fact that Rhino+NoInst uses a more complex noise distribution transformed by a normalizing flow, whereas SCOTCH uses a Gaussian noise modulated by a diffusion matrix.
>
> > "What is the effect of the sparsity prior on the structure? Can a different type of sparsity be used?"
>
> The reason we need the sparsity prior is because in theory, a fully connected graph will always have the greatest capacity to fit data. The purpose of incorporating a sparsity term is to avoid false positive edges, i.e. spurious edges that may slightly improve likelihood but are not present in the underlying system. In real-world systems (e.g. DREAM3), we found this helped to more accurately predict the ground-truth graph.
>
> L1-based sparsity priors (Lasso, group Lasso, adaptive group Lasso) are the most common type of prior used in structure learning, so we adopt this for our method. That said, there are other possibilities. For example, another prior sometimes used in DAG structure learning (Eggeling et al. 2019) assigns probability $p(G) \propto \prod_{d=1}^{D} \frac{1}{{D-1 \choose |pa(d)|}}$, where $pa(d)$ is the set of parents of the $d^{\textnormal{th}}$ variable. In theory we can use any graph prior that assigns positive probability to every graph. The L1 prior turns out to be particularly convenient (in combination with our variational graph posterior) as the KL-divergence in Eq. (14) is tractable.
>
> (Eggeling et al. 2019) Ralf Eggeling, Jussi Viinikka, Aleksis Vuoksenmaa, Mikko Koivisto. On Structure Priors for Learning Bayesian Networks. AISTATS 2019.

---

### Author Response · Authors · 2023-11-17
**Overall Response**

We would like to thank all reviewers for their hard work and thoughtful feedback. We are grateful for the largely positive reception, and recognition of the novelty and contributions of our work. In particular, the reviewers seem to agree that our work introduces novel capabilities for structure learning by leveraging continuous-time modelling:

> "One of the primary strengths of this paper is its originality in tackling the problem of structure learning in continuous-time data..." (`pC52`)

> "Continuous dynamics modeling is a good extension ... because of the ability to incorporate irregularly sampled time series" (`rM8f`)

> "The paper gives an interesting connection between structural learning and SDE" (`XYcj`)

Supporting this, we are glad that the reviewers also praised the strong empirical results of our proposed method, SCOTCH, and our theoretical results on identifiability and consistency.

> "The paper provides a strong theoretical foundation for the proposed method..." (`pC52`)

> "The method empirically performs well in comparison to existing models ... (and) is additionally well motivated with a few theoretical results..." (`gu2c`) \\

We acknowledge also the concerns and questions raised by the reviewers, and have responded individually to each reviewer. Perhaps the main objection is from Reviewer gu2c about the differences of SCOTCH compared to some prior works on latent SDE learning. We would like to emphasize the conceptual contribution to structure learning (which reviewers agreed on), our theoretical results which are new and distinct from prior results, as well as practical differences that preclude prior methods from being directly used for structure learning like SCOTCH.

Finally, we have made changes to the manuscript, which are labelled with the reviewer comment and we are highlighting them in color.

---

> ### Author Response · Authors · 2023-11-23
> **Official Comment by Authors**
>
> As the discussion period comes to a close, we would like to once again thank all reviewers for their feedback, which has been invaluable in enabling us to improve the paper.
>
> Our latest manuscript revision is to conform to the paper length constraints, and includes an additional edit pass on the Appendix as mentioned by reviewer gu2c.

---

### Meta-Review · Area_Chair_qPRR · 2023-12-05

**Metareview:**

The authors tackle the problem of discovering relationships among variables from continuous-time temporal observations. By combining neural SDE with variational inference, their proposed method, SCOTCH, allows learning/predicting observations at arbitrary time points--an important challenge in this area. They show statistical results--sufficient criterion for structural identifiability and consistency--and improved performance on synthetic and real datasets. Both the theory and empirics are strong.

All reviewers gave a positive evaluation after discussion.

**Justification For Why Not Higher Score:**

Stationary dynamics are required; could have stronger experiments and comparison with previous work.

**Justification For Why Not Lower Score:**

All reviewers evaluated the paper positively. The paper has good results on an important problem.

---

### Decision · Program_Chairs · 2024-01-16

Accept (poster)